Subject Category:
Biology (whole organism)

Subject Areas:
palaeontology/evolution/biomechanics

Keywords:
digital reconstruction, finite-element analysis, macroevolution, mammalian evolution, Multituberculata, Rodentia

Authors for correspondence:
Neil F. Adams
e-mail: nfa10@leicester.ac.uk
Emily J. Rayfield
e-mail: e.rayfield@bristol.ac.uk

†Present address: Centre for Palaeobiology Research, School of Geography, Geology and the Environment, University of Leicester, Leicester LE1 7RH, UK.

# Functional tests of the competitive exclusion hypothesis for multituberculate extinction

Neil F. Adams[1,†], Emily J. Rayfield[1], Philip G. Cox[2,3], Samuel N. Cobb[2,3] and Ian J. Corfe[4]

[1]School of Earth Sciences, University of Bristol, Bristol BS8 1RJ, UK
[2]Department of Archaeology, University of York, York YO1 7EP, UK
[3]Centre for Anatomical and Human Sciences, Hull York Medical School, University of York, York YO10 5DD, UK
[4]Institute of Biotechnology, University of Helsinki, 00014 Helsinki, Finland

 NFA, 0000-0003-2539-5531; EJR, 0000-0002-2618-750X; PGC, 0000-0001-9782-2358; SNC, 0000-0002-8360-8024; IJC, 0000-0002-1824-755X

Multituberculate mammals thrived during the Mesozoic, but their diversity declined from the mid-late Paleocene onwards, becoming extinct in the late Eocene. The radiation of superficially similar, eutherian rodents has been linked to multituberculate extinction through competitive exclusion. However, characteristics providing rodents with a supposed competitive advantage are currently unknown and comparative functional tests between the two groups are lacking. Here, a multifaceted approach to craniomandibular biomechanics was taken to test the hypothesis that superior skull function made rodents more effective competitors. Digital models of the skulls of four extant rodents and the Upper Cretaceous multituberculate *Kryptobaatar* were constructed and used (i) in finite-element analysis to study feeding-induced stresses, (ii) to calculate metrics of bite force production and (iii) to determine mechanical resistances to bending and torsional forces. Rodents exhibit higher craniomandibular stresses and lower resistances to bending and torsion than the multituberculate, apparently refuting the competitive exclusion hypothesis. However, rodents optimize bite force production at the expense of higher skull stress and we argue that this is likely to have been more functionally and selectively important. Our results therefore provide the first functional lines of evidence for potential reasons behind the decline of multituberculates in the changing environments of the Paleogene.

# 1. Introduction

The Multituberculata (Allotheria, Mammalia) was one of the most successful orders of mammals to have ever lived, persisting for over 130 Myr, from the Middle Jurassic [1] to the late Eocene [2,3]. The extinction of this successful group has remained a long unresolved issue. Multituberculates survived the Cretaceous–Paleogene mass extinction (when *ca* 75% of all species died out [4]) to achieve peak global diversity in the early Paleocene [5], but this diversity was gradually eroded through the Paleogene leading to their eventual extinction.

Although environmental changes and the radiation of potential predators (such as strigiform raptors and creodont and carnivoran mammals) have been proposed as explanations for multituberculate extinction [6], competitive exclusion by the morphologically convergent rodents (figures 1 and 2) is the most commonly cited cause [6,20,21]. The competitive exclusion hypothesis (hereafter referred to as the 'CE hypothesis') is currently based on (i) negatively correlated (double wedge) diversity patterns and (ii) overlap in proposed ecological niche occupation, with similarities in reconstructed diet, body size, diurnal behaviour, locomotion and habitat [21]. However, studies going beyond simple diversity correlations and niche reconstructions to quantitatively demonstrate the competitive superiority of rodents are lacking.

The wider role of competition in driving macroevolutionary processes (such as extinction) also remains unclear [6], with the study of competitive exclusion having previously been neglected and even denigrated [22]. Many cases from the fossil record where a group was suggested to have been outcompeted by another (e.g. brachiopods by bivalves, synapsids by archosaurs) have been suggested to more likely result from mass extinction effects and opportunistic replacement [23]. Nevertheless, recent studies have increasingly supported interspecific competition as a genuine agent in macroevolution [24–26]. Benton [23] believed the evidence for competitive exclusion of multituberculates by rodents was still open to interpretation, and Jablonski ([6], p. 724) suggested that the case for multituberculate exclusion was strong but deserved 'more detailed evaluation of spatial patterns, dynamics of subclades, and morphological and functional spaces occupied by the potential interactors'. If multituberculate extinction was due to competition from rodents, it is currently unclear which characteristics provided rodents with a competitive advantage since explicit functional comparisons between the groups have not been attempted.

In this paper, we employ methods in computational biomechanics to assess the CE hypothesis by providing the first functional comparisons between multituberculates and rodents to test one aspect of competitive superiority. Our study focuses on one representative of multituberculate skull morphology (see §4.3 for a discussion of this limited sample) as a first attempt to demonstrate the suitability of functional analyses to test hypotheses of competitive exclusion. Given the craniomandibular convergence and the inferred similarity in diet, we hypothesized that if the evolution of the rodent craniomandibular apparatus allowed for more efficient procurement and processing of the same food items, multituberculates would have been left at a selective disadvantage. As such, we aimed to establish whether the anatomy and function of rodent skulls are superior to those of multituberculates. Here, function is assessed through the mechanical performance of the skull (the cranium and mandible). Three hypotheses were designed and tested to achieve this aim:

(1) The morphology of rodent skulls is better able to deal with stresses imposed during multiple feeding behaviours (incisor gnawing and molar chewing), and consequently rodent skulls exhibit lower stresses than those of multituberculates. This hypothesis was tested by conducting finite-element analysis (FEA [27,28]) on digital models of multituberculate and rodent skulls to examine feeding-induced stress distributions.

(2) Rodent skull morphology enables the generation of higher bite forces than multituberculates, thereby facilitating access to a wider range of food resources. To test this, bite force metrics (mechanical efficiency [18] and bite force quotient [29]) were calculated from FEA outputs.

(3) Rodent skull morphology permits greater resistance to bending and torsional forces, enabling more efficient processing of resistant food items than multituberculates. Principles from cantilever beam theory were applied to the digital skull models to calculate resistances to bending and torsion (e.g. [30,31]).

# 2. Material and methods

## 2.1. Sample

The species *Kryptobaatar dashzevegi* Kielan-Jaworowska 1970 was chosen as a representative of multituberculate skull morphology, the reasons for which are fourfold. Firstly, *K. dashzevegi* is represented

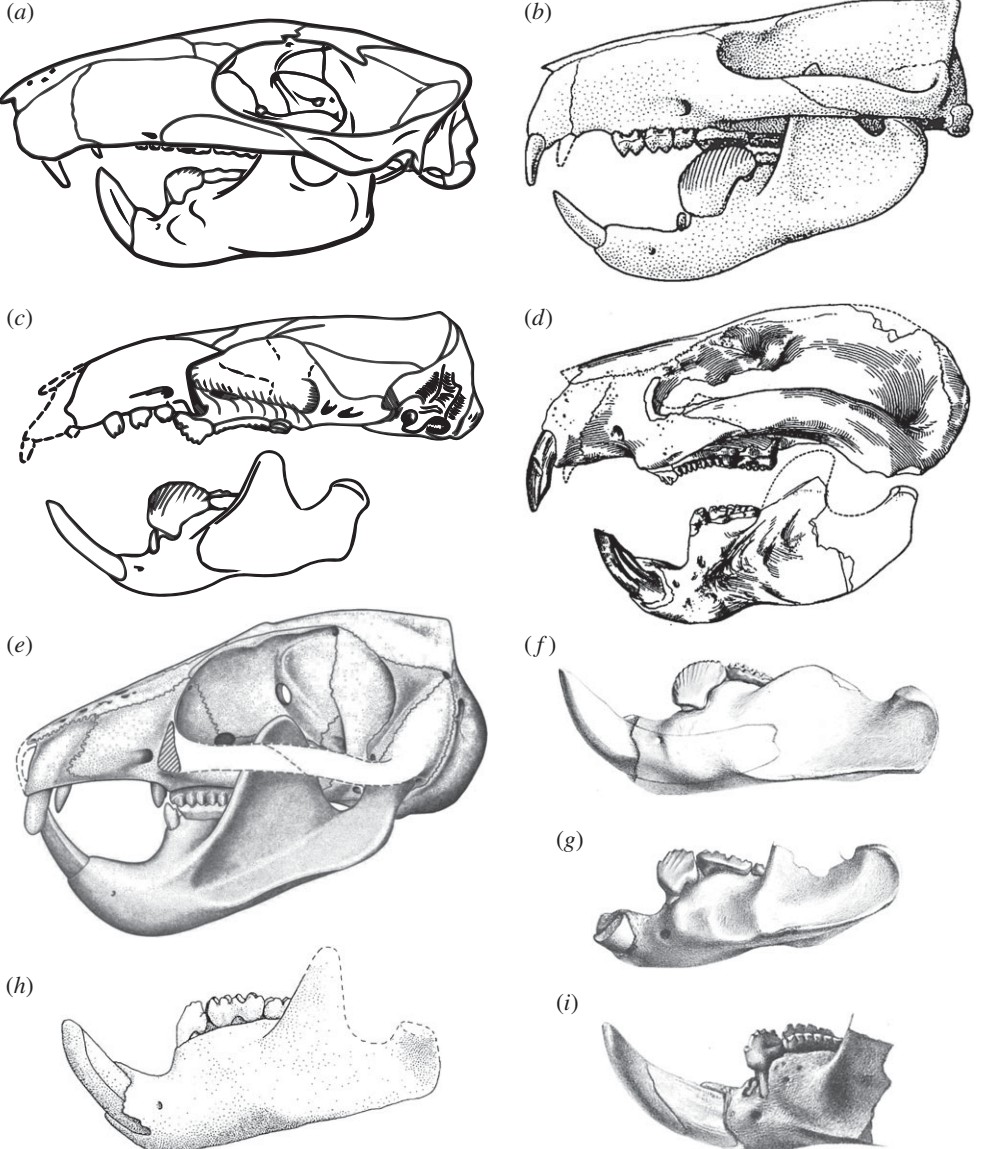

**Figure 1.** Skull of the Upper Cretaceous multituberculate *Kryptobaatar dashzevegi* (Djadochtatheriidae) (*a*) for comparison with the crania and hemimandibles of representatives of the main Paleogene multituberculate families (*b*–*i*). Relatively complete skulls, or reconstructions made from composite specimens, are known for (*b*) *Ptilodus montanus* (Ptilodontidae), (*c*) *Ectypodus tardus* (Neoplagiaulacidae), (*d*) *Taeniolabis taoensis* (Taeniolabididae) and (*e*) *Lambdopsalis bulla* (Lambdopsalidae). Many more specimens are known from less complete fossils, including isolated hemimandibles of (*f*) *Stygimys kuszmauli* (= *Eucosmodon gratus*; Eucosmodontidae), (*g*) *Pentacosmodon pronus* (Microcosmodontidae), (*h*) *Catopsalis foliatus* (Taeniolabididae) and (*i*) *Microcosmodon conus* (Microcosmodontidae). Images in: (*a*) modified from [7], courtesy of the American Museum of Natural History; (*b*) modified from [8]; (*c*) modified from [9] (cranium) and [10] (mandible); (*d*) reproduced from [11], courtesy of the American Museum of Natural History; (*e*) modified from [12], courtesy of the University of Wyoming; (*f*), (*g*) and (*i*) reproduced from [13], courtesy of the American Philosophical Society and (*h*) reproduced from [14].

by more relatively complete skulls than any other Mesozoic mammal and its craniomandibular anatomy has been described and reconstructed in detail from numerous specimens [7,32]. Secondly, a specimen (PSS-MAE 101) collected from the Ukhaa Tolgod locality [33] preserves an anterior, fully articulated skeleton with a skull in near-pristine condition. CT scan data of this specimen are openly available through the Digital Morphology online library (DigiMorph; http://www.digimorph.org/specimens/ Kryptobaatar_dashzevegi), and this is currently the only freely available three-dimensional scan of a well-preserved multituberculate skull. Thirdly, *K. dashzevegi* is a member of an important suborder of multituberculates: the Cimolodonta, which existed from the Lower Cretaceous to the Eocene and were among the multituberculates to co-exist with rodents during the Paleogene. Although *K. dashzevegi* and

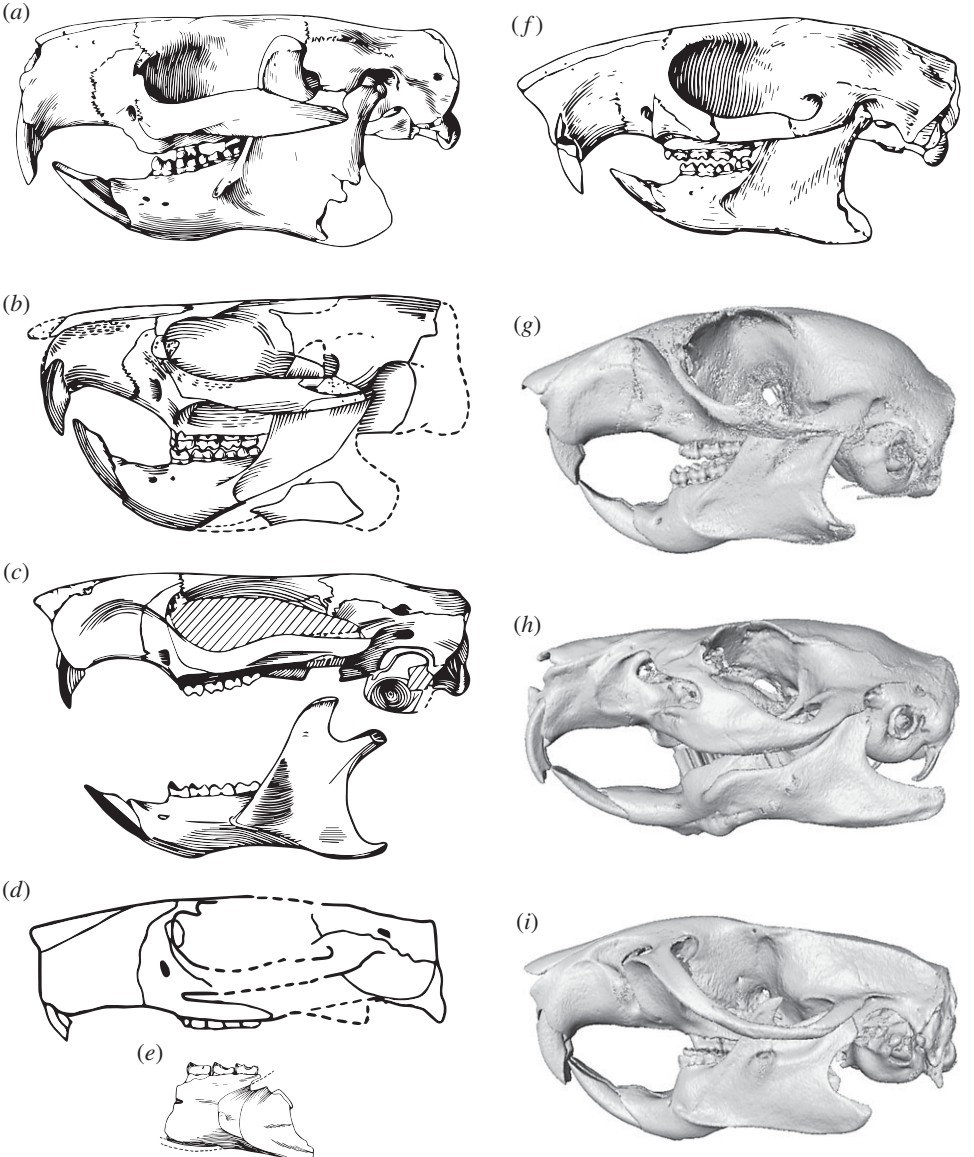

**Figure 2.** Skulls of Paleogene rodents (*a*–*f*) for comparison with the extant rodents used in this study (*g*–*i*). Skulls, or reconstructions from composite specimens, of ischyromyid rodents are known for several Paleogene taxa such as (*a*) *Paramys delicatus*, (*b*) *Thisbemys corrugatus*, (*c*) *Reithroparamys delicatissimus*, (*d*) *Franimys amherstensis*, and (*e*) *Microparamys minutus*, as well as the early sciuravid rodent (*f*) *Sciuravus nitidus*. Digital models of the modern rodents shown are (*g*) *Sciurus carolinensis*, (*h*) *Cavia porcellus* and (*i*) *Rattus norvegicus*. Images in (*a*) and (*f*) modified from [15], courtesy of the American Museum of Natural History; (*b*), (*c*) *and* (*e*) modified from [16], courtesy of the American Philosophical Society; (*d*) modified from [17], courtesy of Springer Nature; (*g*), (*h*) and (*i*) created using CT scan data from [18,19].

the superfamily to which it belongs (the Djadochtatherioidea) are currently only known from the Upper Cretaceous, the species shares multiple craniodental characteristics with other cimolodontans (figure 1) that do survive into the Paleogene [34]. Lastly, the masticatory musculature of the closely related djadochtatherioidean *Nemegtbaatar* Kielan-Jaworowska 1974 has been thoroughly reconstructed based on well-preserved skulls with muscle attachment scars, depressions and ridges [35], which are also readily identifiable on PSS-MAE 101.

The majority of the Paleogene rodent fossil record comprises fragmentary material and isolated teeth, but near-complete skulls, especially from North America (e.g. *Paramys* and *Ischyromys*), are also known [17]. Although several Paleogene rodent skulls have now been scanned using X-ray tomography (e.g. [36]), the scans are not yet openly available. Despite differences in the origins of the masseter muscles between living rodents and their earliest ancestors [18,37], use of extant rodents is justifiable to test the CE hypothesis since many other craniomandibular characteristics typical of rodents are shared and

little changed between extant and extinct forms [17,38] (figure 2). The few differences between extant and extinct forms mainly relate to the protrogomorph condition of the latter, with the anterior margin of the masseter extending further anteriorly along the mandible and further anteriorly along the zygomatic arch, and even on to the rostrum, in extant taxa. In addition, extinct Paleogene rodents, such as *Paramys*, have a broader and better-developed coronoid process, a less well-developed mandibular angle and more primitive molar crown patterns [39]. However, there are many similarities including, for example, the dorsoventrally low nature of the skull with essentially parallel dorsal and ventral margins, the anterior position of the well-developed zygomatic arches and an elongated rostrum, the narrowing of the skull in the postorbital region, the anteroposterior elongation of the glenoid fossa and mandibular condyle, a long diastema between incisor and cheek teeth, and a posteriorly curving coronoid process that lies well above the plane of the molar row [17,39]. Additionally, extant rodents have been the focus of recent functional analyses [18,40], which can be directly compared with similar analyses of multituberculate skulls.

We therefore compared the craniomandibular function of *Kryptobaatar* with four extant rodent species representing the three major morphotypes of rodent masticatory muscle configuration. Skulls of *Sciurus carolinensis* (grey squirrel), *Cavia porcellus* (guinea pig) and *Rattus norvegicus* (brown rat), representing the sciuromorph, hystricomorph and myomorph muscle morphotypes, respectively, have previously been subjected to cranial functional analysis [18]. Contrast-enhanced CT scans of these specimens were acquired by Cox & Jeffery [19] and were used here (see §2.2). CT scans of *Mus musculus* (specimen TMM M-3196 obtained from the DigiMorph online library), another representative of the myomorph condition, were used to produce a house mouse model. Muscle data for the mouse model were obtained from the contrast-enhanced CT scan of *M. musculus* previously used to describe the morphology of the mouse masticatory musculature [41]. CT scans of an additional *R. norvegicus* (specimen TMM M-2272 obtained from the DigiMorph online library) were also used. Further details of the sample are provided in the electronic supplementary material.

## 2.2. Model creation

Digital models of the crania, hemimandibles and masticatory musculature required for FEA were created from CT scan data in Avizo Lite 9.2.0 (FEI Visualization Sciences Group, Bordeaux, France). The digital models and results from cranial FEA of three extant rodents (*Sciurus*, *Cavia* and *Rattus*) were taken directly from the Cox *et al.* [18] study and were used for comparison with the new results for the *Kryptobaatar* cranium calculated herein. The digital model of the *Mus* cranium was created from the DigiMorph CT scan data as part of this study (since *Mus* was not included in previous FEA work), and new hemimandibular models were made for *Kryptobaatar* and all rodent taxa (since previous work only focused on rodent crania) using existing CT scans. Where bones of the *Kryptobaatar* skull were incompletely preserved they were reconstructed with reference to other specimens (e.g. PSS-MAE 113) and a skull reconstruction based on all available material [7].

Muscle forces for the guinea pig, rat and squirrel were used from [18], and those for the mouse and *Kryptobaatar* were calculated using the 'dry skull' method [42], which uses muscle physiological cross-sectional area to determine force (for further details, see electronic supplementary material).

## 2.3. Finite-element analysis

Digital models of hemimandibles and crania were converted into three-dimensional meshes using HyperMesh 14.0 (Altair Engineering Inc., Troy, MI, USA). Mesh components were modelled as linearly elastic and isotropic, homogeneous material properties were applied to the components based on measured values from rodents [18]. Sensitivity tests varying material properties revealed minimal differences in output results (see electronic supplementary material).

Cranial models were constrained at the temporomandibular joints (TMJs) and at the biting teeth. Hemimandibular models were constrained on the condyle at the TMJ, at the mandibular symphysis and at the biting tooth. Masticatory muscle forces for each taxon were used to load the models. The force for each muscle was divided evenly over a number of nodes across the origin and attachment sites. Vectors were created between these sites to provide load orientations (e.g. figure 3). Further details of model constraints and loads are provided in the electronic supplementary material.

FE models were solved in Abaqus 6.14–1 (Dassault Systèmes Simulia Corp., Providence, RI, USA) and the von Mises stress of each element was extracted from the model output and plotted as a

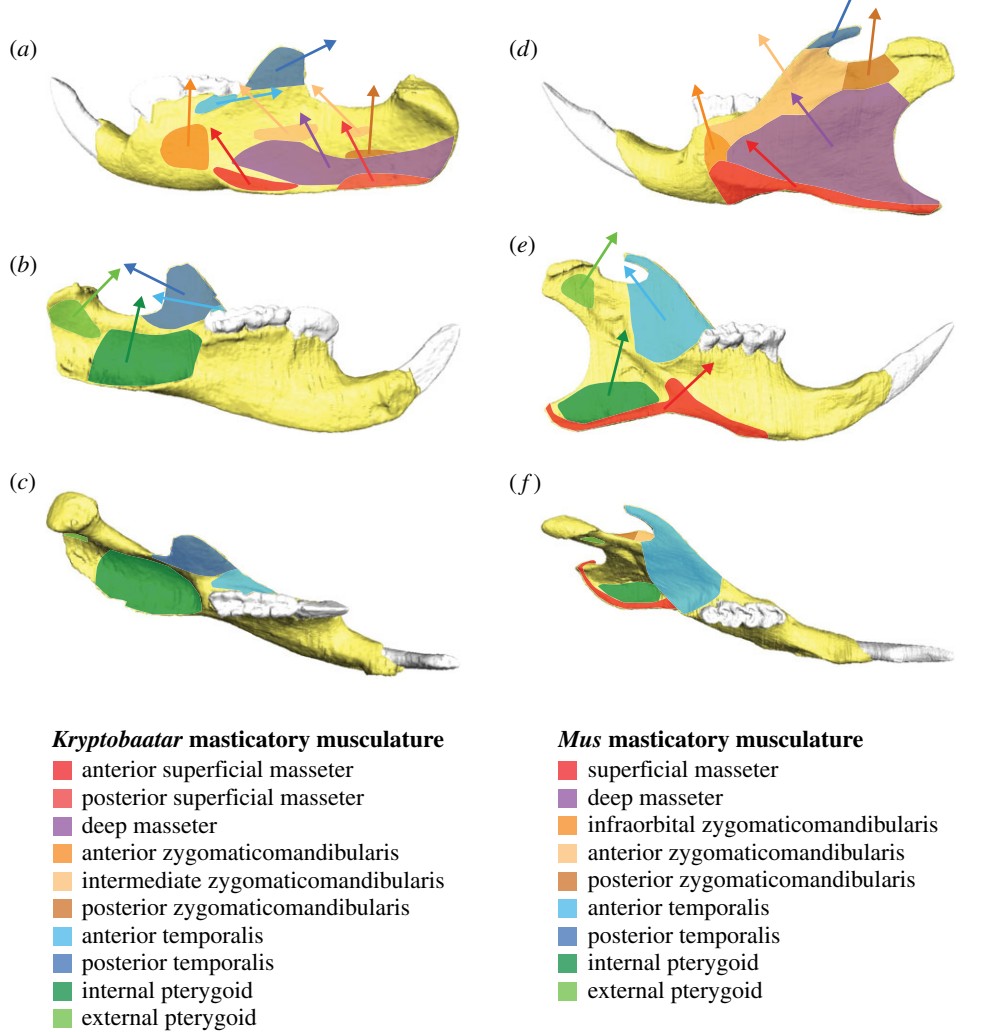

**Figure 3.** Muscle attachment sites and vectors on the hemimandibles of (*a–c*) *Kryptobaatar* and (*d–f*) *Mus*. (*a*) and (*d*) in lateral view, (*b*) and (*e*) in medial view, (*c*) and (*f*) in occlusal/dorsal view. Attachment sites are based on [35] and [41]. Muscle vector orientations were calculated during finite-element model construction in HyperMesh software (see Material and methods).

coloured contour map to show stress distributions. Median stress was calculated for each model, since mean stress is often affected by erroneously high values near point constraints [28,43].

## 2.4. Bite force metrics

The mechanical efficiency (ME) of biting for each model was determined by dividing the estimated bite force by the total applied muscle force [18], which indicates how well muscle force is converted into biting force. To quantify the estimated bite force, reaction forces at each tooth constraint in the FE model were summed.

To compare output bite forces from FEA, the size-corrected bite force quotient (BFQ) [29] was calculated for all taxa at incisor and rearmost molar bites. BFQ is a bite force metric that uses body mass to take the size of organisms into account when comparing bite forces. The equation used to calculate BFQ is provided in the electronic supplementary material. BFQ was also calculated for biting at the lower fourth premolar (p4) for *Kryptobaatar*, since multituberculates are thought to have used their plagiaulacoid premolars rather than, or in addition to, their incisors to break into tough food items [44,45].

## 2.5. Beam biomechanics

Cantilever beam theory has been used to study the resistance of mandibles and cranial rostra to bending and torsion for a wide range of taxa (e.g. [30,31]). Strength in bending was determined by the section

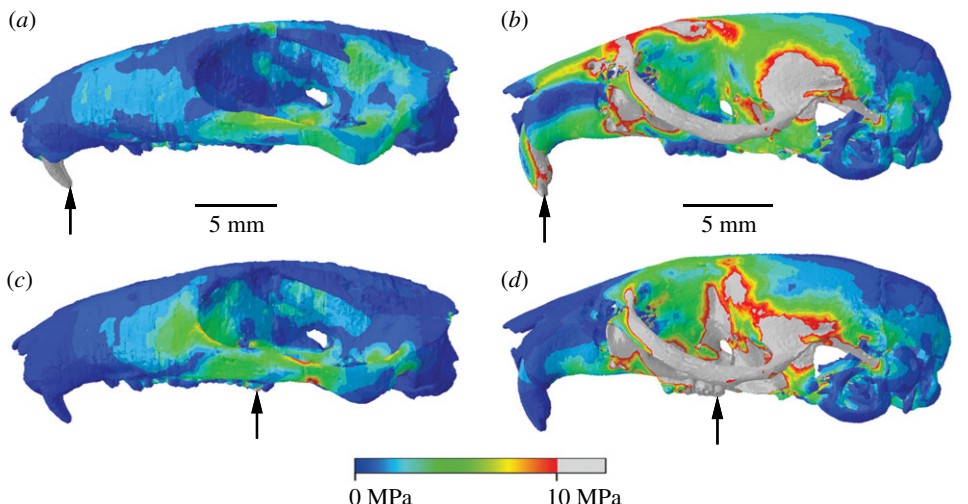

**Figure 4.** Cranial stress distributions during incisor biting (*a,b*) and rearmost molar biting (*c,d*) for *Kryptobaatar* (*a,b*) and *Mus* (*c,d*). Grey areas indicate von Mises stresses greater than 10 MPa. Arrows indicate the biting tooth. Scales in (*a*) and (*b*) apply to (*c*) and (*d*), respectively.

modulus ($Z$), which was calculated by dividing the second moment of area ($I$) by the distance between the centroidal axis and outer edge of the object in the plane of bending. Following [31], the section modulus was measured in the dorsoventral ($Z_x$) and mediolateral ($Z_y$) planes to calculate resistance to bending in each orientation. The ability of a beam to resist torsion is provided by the polar moment of inertia ($J$), which is the sum of the dorsoventral ($I_x$) and mediolateral ($I_y$) second moments of area.

Coronal cross-sectional image stacks through the three-dimensional digital models of the crania and hemimandibles were imported into the plugin MomentMacro v1.4B (http://www.hopkinsmedicine.org/fae/mmacro.html) for ImageJ v1.51 h (National Institutes of Health, Bethesda, MD, USA) to calculate section moduli and polar moments of inertia. Calculations were performed once every 20 slices along the hemimandibles and rostra for the rat, mouse, guinea pig and squirrel image stacks and once every 10 slices for the *Kryptobaatar* stacks due to the lower resolution of the *Kryptobaatar* scan. To correct for size discrepancies and to simply evaluate the effect of morphology, each model was scaled to the same length.

Pairwise Wilcoxon signed-rank tests were performed in SPSS Statistics 23 (IBM Corp., Armonk, NY, USA) to test for significant differences in resistance to bending and torsion for the scaled data at equivalent positions along the hemimandibles and rostra. Bonferroni *p*-value corrections were made to account for the increased likelihood of type I statistical errors resulting from multiple comparisons.

# 3. Results

## 3.1. Stress distribution

Comparison of von Mises stress distributions across the cranium reveals that *Kryptobaatar* was under lower stress overall than the mouse at both incisor and molar bites, with only very small areas exhibiting stresses greater than 10 MPa (figure 4). Despite the differences in magnitude, the areas under highest and lowest stresses were similar for both taxa (figure 4). The biting tooth and zygomatic arches, ventral surface of the maxilla and anterior basicranium all experienced relatively high stress. The rostrum, posterior basicranium, posterior parietals, occipital region and auditory bullae experienced low stress in all cases. The dorsal surface of the mouse cranium, notably the frontal bones and frontomaxillary sutures, was under much higher stress than *Kryptobaatar* in all biting scenarios, and stress across the dorsal surface in *Kryptobaatar* was negligible during distal molar biting. As would be expected, the rostrum was more stressed during incisor biting than molar biting due to a greater bending moment in the skull, and the area of higher stress shifted from the anterior to posterior portion of the maxilla between incisor and molar bites.

The mouse similarly exhibited much higher stresses than *Kryptobaatar* in the hemimandibular FE models (figure 5). During incisor biting, the majority of the mouse hemimandible experienced stresses

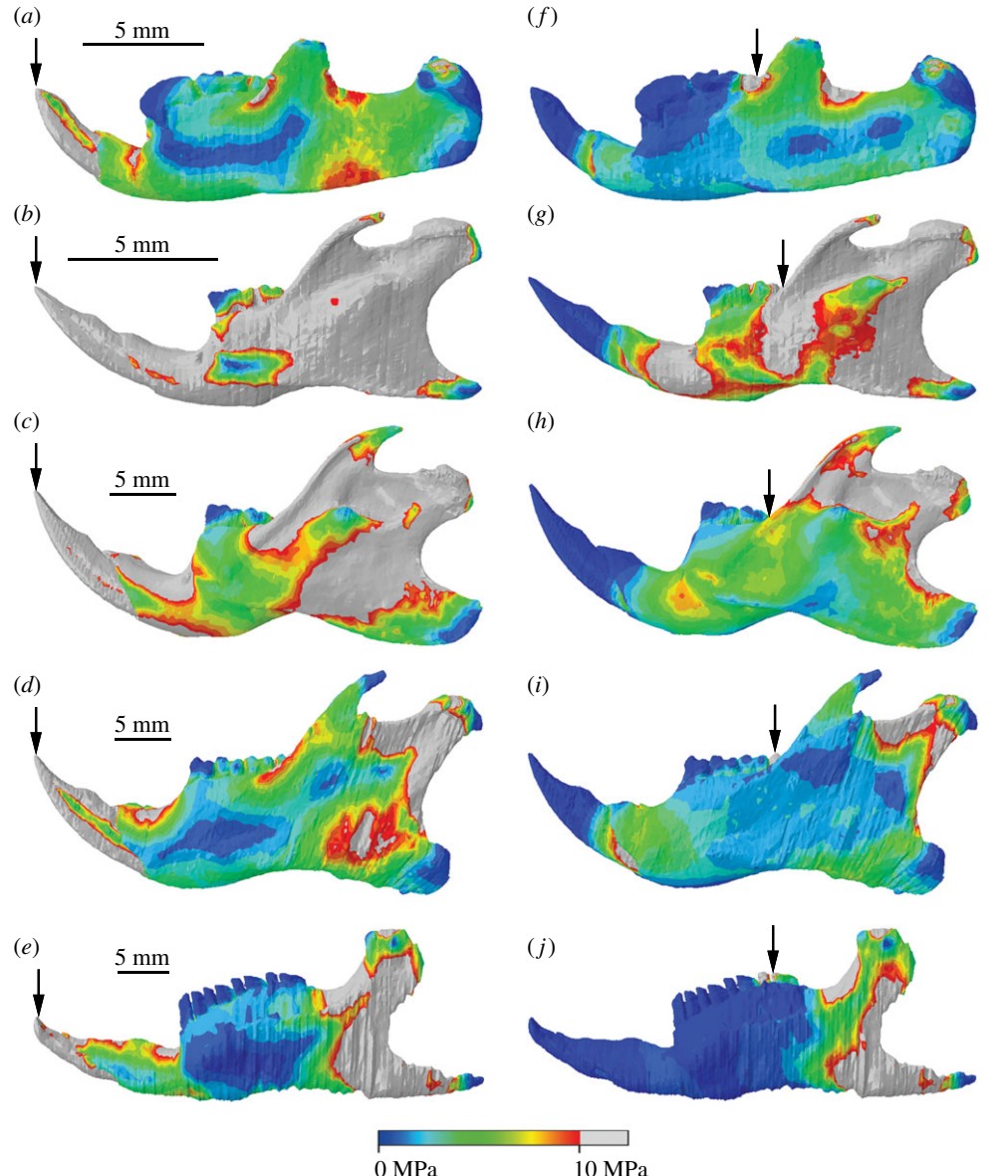

**Figure 5.** Hemimandibular stress distributions during incisor biting (*a*−*e*) and rearmost molar biting (*f*−*j*) for *Kryptobaatar* (*a, f*), *Mus* (*b, g*), *Rattus* (*c, h*), *Sciurus* (*d, i*) and *Cavia* (*e, j*). Grey areas indicate von Mises stresses greater than 10 MPa. Arrows indicate the biting tooth. Scales in (*a*−*e*) apply to (*f*−*j*).

greater than 10 MPa, along the incisor tooth, dorsal and ventral surfaces of the bone along the diastema, the mandibular ramus, coronoid process and condyle (figure 5*b*). Only the molar teeth, the posterior-most region of the mandibular angle and thin regions along the lateral surfaces of the alveolar region were under relatively low stresses. Areas of high stress during incisor biting on the *Kryptobaatar* hemimandible were limited to the incisor tooth, dorsal surface of the bone along the diastema, a small area immediately posterior to the coronoid process and the dorsal surface of the TMJ (figure 5*a*). The areas surrounding the symphyseal point constraints were areas of high stress in all hemimandibular models, and during molar biting the symphyseal point constraints acted to extend regions of higher stress anteriorly along the hemimandible.

Stress distributions across the rat hemimandible were largely similar to the mouse during incisor biting, but overall stresses were lower, especially on the lateral surfaces of the alveolar region (figure 5*c*). The squirrel and guinea pig hemimandibles had the smallest areas of high stress among the rodents (figure 5*d,e,i,j*) and were more comparable to stress distributions seen in *Kryptobaatar*. The largest area of high stress in these taxa was seen in the area between the coronoid process and condyle, and from the condyle down towards the angle. All hemimandibular models showed reduced

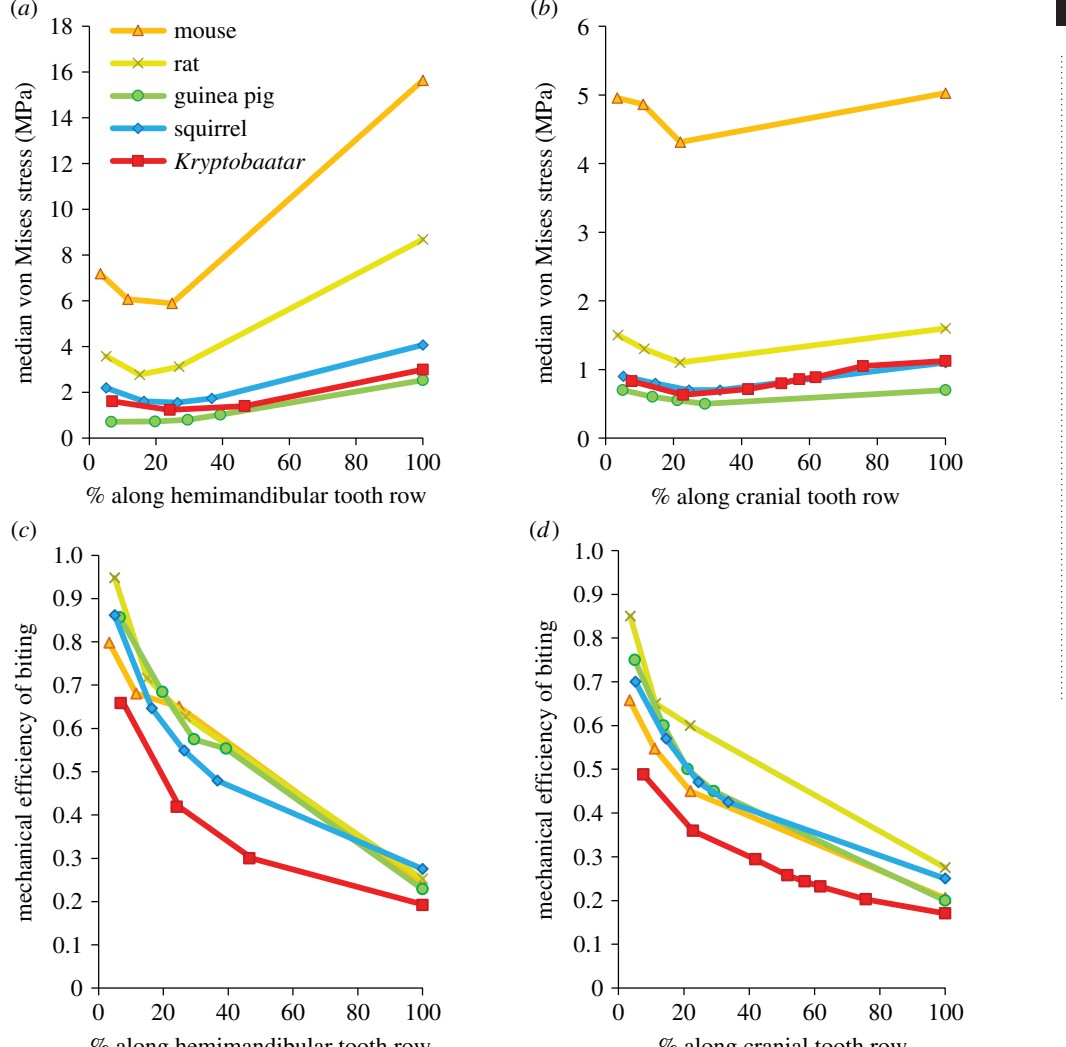

**Figure 6.** Median von Mises stress (*a,b*) and mechanical efficiency (ME) (*c,d*) of the hemimandibles and crania at different bite points for rodents and *Kryptobaatar*. In each case, 0% along the tooth row corresponds to the back of the rearmost molar and 100% corresponds to the incisor tip. Cranial stress and cranial ME values for the rat, guinea pig and squirrel were taken directly from [18]. Key in (*a*) also applies to (*b*−*d*).

areas of high stress during molar biting compared with incisor biting (figure 5). In all taxa, the highest stresses were found between the coronoid process and condyle, with reduced stress over the lateral surfaces of the alveolar region and ramus compared to incisor biting.

## 3.2. Average stress

Trends established from stress distribution results were also borne out in measurements of median stress averaged across all model elements. It is clear that the mouse and rat experienced the highest average stresses in both hemimandibular and cranial models (figure 6*a,b*). In all models, highest stresses were seen during incisor and distal molar biting, with stress distributions shifting anteriorly and posteriorly in each case. Lowest average stresses were experienced during first or second molar biting. *Kryptobaatar* fell between the lower average values of the squirrel and guinea pig, confirming visual similarities in stress distribution across the hemimandibles (figures 5 and 6*a,b*).

## 3.3. Mechanical efficiency and BFQ

As well as producing absolutely higher bite forces, as predicted by reaction forces in FE models (see electronic supplementary material), the size-independent metric of mechanical efficiency showed that

**Table 1.** Values of body mass, from the Animal Ageing and Longevity Database (http://genomics.senescence.info/species) for extant rodents and from [32] for *Kryptobaatar*, and bite forces from FE model outputs used to calculate the size-corrected bite force quotient (BFQ) at the incisors (BFQ$_i$), rearmost molars (BFQ$_m$) and fourth lower premolar for *Kryptobaatar* (BFQ$_p$). Equation used for BFQ calculation is provided in the electronic supplementary material.

| | body mass, BoM (kg) | incisor bite force, IB$_s$ (N) | BFQ$_i$ | molar bite force, MB$_s$ (N) | BFQ$_m$ | p4 bite force PB$_s$ (N) | BFQ$_p$ |
|---|---|---|---|---|---|---|---|
| *Kryptobaatar* | 0.0853 | 2.51 | 21 | 8.59 | 73 | 3.92 | 33 |
| *Mus* | 0.0205 | 5.05 | 101 | 16.71 | 335 | | |
| *Rattus* | 0.300 | 11.84 | 47 | 44.48 | 177 | | |
| *Cavia* | 0.728 | 10.08 | 24 | 37.80 | 88 | | |
| *Sciurus* | 0.533 | 15.19 | 43 | 47.55 | 134 | | |

rodents could better transfer muscle input force into biting output force than *Kryptobaatar* (figure 6*c,d*). For hemimandibular models, the percentage difference in ME was 23–28% higher among rodents. In cranial models, percentage difference was also higher but more variable at 14–42% (electronic supplementary material, table S24).

The size-corrected BFQ confirmed the ME results, with all rodents showing higher BFQ values at both incisor and molar biting than *Kryptobaatar* (table 1). The BFQ of *Kryptobaatar* during p4 biting was also lower than all incisor BFQs of rodents, except for the guinea pig, which is a specialist grazer adapted for molar chewing rather than incisor gnawing (table 1). BFQ values for *Kryptobaatar* incisor and molar biting were closest to the guinea pig and were 13% and 19% lower, respectively.

## 3.4. Resistance to bending and torsion

Statistical comparisons of bending resistances reveal that the squirrel was the only rodent significantly different from *Kryptobaatar* in dorsoventral bending of the hemimandible ($Z = -2.068$, $p = 0.039$; see electronic supplementary material, table S27), but even this difference shifted to non-significance after applying the Bonferroni correction. The only difference to remain significant after the Bonferroni correction was between the rat and *Kryptobaatar* in mediolateral bending ($Z = -2.746$, $p = 0.006$), with the rat having significantly lower resistance. Overall, these results suggest there was little difference in hemimandibular resistance to dorsoventral or mediolateral bending between rodents and *Kryptobaatar* (figure 7*a,b*).

However, significant differences were found between *Kryptobaatar* and rodents in all comparisons for resistance along the rostrum (see electronic supplementary material, table S27), except in two cases (and in the latter only after applying the Bonferroni correction): *Kryptobaatar* was not significantly different from the guinea pig in mediolateral bending resistance ($Z = -0.087$, $p = 0.931$) and in torsional resistance ($Z = -2.172$, $p = 0.030$). In all other cases, the squirrel and guinea pig had significantly higher resistance compared with *Kryptobaatar*, and the mouse and both rat models significantly lower (figure 7*d,e*).

Since resistance to torsion ($J$) is simply the sum of the second moments of area in the dorsoventral ($I_x$) and mediolateral ($I_y$) axes, and because $I_x$ was larger than $I_y$ in most cases, patterns of torsional resistance were largely similar to the dorsoventral bending resistances (figure 7*a–f*).

# 4. Discussion

## 4.1. Mixed support for the competitive exclusion hypothesis

Results from functional tests provide apparently conflicting evidence for the CE hypothesis. Masticatory stress and skull strength results refute, while bite force results support, our initial hypotheses (see Introduction).

The skull (both cranium and hemimandible) of *Kryptobaatar* was not under higher stress than the extant rodents, in fact only the hystricomorph (guinea pig) produced consistently lower average stresses. The multituberculate skull was therefore better able to withstand feeding-induced stresses

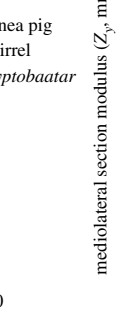
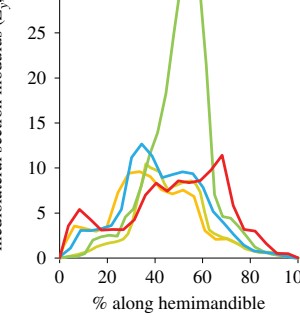
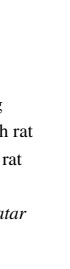
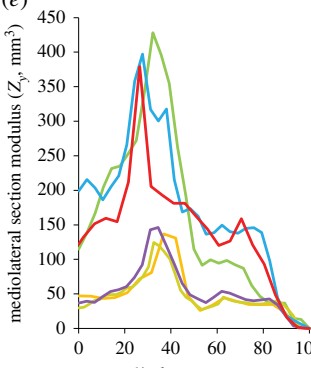
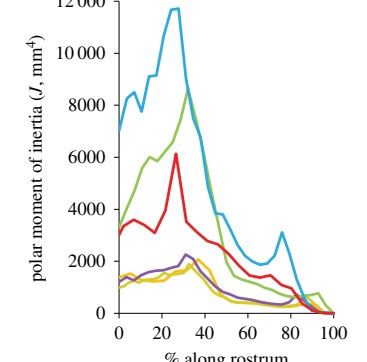

**Figure 7.** Resistance of the scaled hemimandibles (*a*–*c*) and rostra (*d*–*f*) to dorsoventral (*a*,*d*) and mediolateral (*b*,*e*) bending and torsion (*c*,*f*). In (*a*–*c*), 0% corresponds to the posterior of the mandibular angle and 100% corresponds to the incisor tip. In (*d*–*f*), 0% corresponds to the back of the rearmost upper molar and 100% corresponds to the anterior of the rostrum. Keys in (*a*) and (*d*) also apply to (*b*,*c*) and (*e*,*f*), respectively.

than several extant rodents, refuting our first hypothesis. Similarly, the myomorph (mouse and rat) rostra showed significantly less resistance to bending and torsion than that of the multituberculate with only the sciuromorph (squirrel) having significantly higher resistance. Analysis of the hemimandibles revealed minimal differences in bending and torsion resistance between rodents and multituberculates. Together these results refute our third hypothesis. Support for the CE hypothesis is only found from comparison of bite force metrics, with higher ME and BFQ values among all rodents, as predicted by our second hypothesis. Despite this mixed support, if these patterns extend to Paleogene rodents and multituberculates more broadly, optimization for higher bite force at the cost of higher stress may have given rodents a selective advantage over multituberculates, providing tentative support for a model of competitive exclusion.

## 4.2. Masticatory and musculoskeletal differences between multituberculates and rodents

Differences in stress distributions between multituberculate and rodent FE models might be expected because, beyond broad convergences in craniodental anatomy, there are differences in the arrangement of muscles and bones of the skull and in resultant masticatory processes between groups. The most notable difference is in the opposing direction of masticatory power strokes, which in turn is linked to the majority of differences in cranial musculoskeletal anatomy between multituberculates and rodents. During the multituberculate power stroke of molar grinding, the mandible was retracted (palinal motion) with the molars in tight occlusion and there was minimal lateral movement [44,46]. Conversely, in rodents the power stroke is protractory (proal motion) with variable transverse components [47,48]. The proportions of muscles required in forward and backward movement of the mandible are therefore quite different. The temporalis muscles (especially the lateral temporalis, also known as the posterior temporalis) are particularly well developed in multituberculates, which has been related to mandible retraction during molar grinding [46]. In rodents, with a proal power stroke, the masseter muscles are especially dominant and the temporalis muscles make up lower proportions of total adductor

musculature [19] (see electronic supplementary material). In FE models, these different muscle configurations have subtle effects on stress patterns. For example, areas of temporalis muscle attachment around the coronoid process in hemimandibular models of *Kryptobaatar* are particularly highly stressed. By comparison, areas of masseter muscle attachment on the ascending ramus in hemimandibular models are among the most stressed areas in rodent models.

Morphological differences in the glenoid fossa, condylar and coronoid processes, angular process and pterygoid shelf have also been linked to these opposing jaw movements and are discussed at length by Gambaryan & Kielan-Jaworowska [35]. To summarize, the glenoid fossa is posteroventrally sloping in multituberculates to allow the upper and lower molar row to occlude during the power stroke; the opposite condition is required in rodents for molar occlusion. In multituberculates, the posteriorly sloping condylar process contributes to the convex posterior margin of the mandible with no angular process, while in many rodents the posterior margin of the mandible is concave, curving underneath a prominent condylar process before extending posteriorly to form a prominent angular process. In rodents and other therian mammals, the angular process is an important attachment site for muscles, such as the superficial masseter and the internal pterygoid (also known as the medial pterygoid), and is suggested to have evolved to increase mechanical advantage during transverse jaw movements [49]. Although multituberculates lack an angular process, apparently due to the palinal power stroke [35], they possess an extensive pterygoid shelf that makes up all of the ventral margin of the medial side of the mandible posterior to the lower tooth row. This shelf is the attachment site of the internal pterygoid in multituberculates, so is homologous to the therian angular process in terms of muscle attachment. These masticatory and musculoskeletal differences between multituberculate and rodent taxa clearly have consequences for how stresses are distributed across the skulls, but by examining average stresses across all elements of FE models in addition to stress distribution maps, it is possible to directly compare overall stress states of taxa. In addition, metrics such as ME take into account the actions of all masticatory muscles, so results are comparable despite differences between individual muscles.

## 4.3. Caveats to data interpretation

FE models are, by necessity, simplifications of complex biological systems and materials. For example, in reality bones and teeth are not isotropic and homogeneous materials, all masticatory muscles are unlikely to contract simultaneously, and crania are not single units but contain multiple sutured bones [28]. However, FEA can prove very informative when an inductive, comparative approach is taken, the same input assumptions are made between models, and comparative rather than absolute results are of interest [28]. Therefore, this study closely followed the methodology of Cox *et al.* [18] since similar organisms were investigated and their results were ground-truthed in validation experiments.

Compared to the simplifications of FE models, a more likely source of uncertainty comes from the muscle input forces calculated for *Kryptobaatar*. Three-dimensional muscle volumes are known for extant rodents, but reconstructing these for fossil taxa lacking soft tissues is inherently problematic. If the real masticatory muscles of *Kryptobaatar* differed from our reconstructions that are based on osteological correlates, then the reconstructed muscle forces used as FEA inputs could be inaccurate. However, these differences are not detectable when only bones are preserved and we took an approach to minimize uninformed inference in muscle reconstructions, following previous work on the reconstruction of masticatory muscles for other fossil taxa [50–52] (see electronic supplementary material).

It should be noted that the differences in ME identified in this study between the extant rodents and *Kryptobataar* apply to the force produced at the location of specific teeth, and do not take into account tooth–food–tooth contact. This is a potentially important point because the ME of food reduction is further modulated by the occlusal morphology of teeth themselves. For example, the same force applied to a sharp versus a blunt cusp will produce different pressures at tooth–food–tooth contact, due to the size of the area of contact; occlusal morphology can further vary by the number and variation in these cusps at a bite point (dental complexity). Given the variation in occlusal morphology both within and between rodents [53] and multituberculates [5], this is a question worth returning to in future studies.

Additionally, the four extant rodents studied here represent the product of almost 60 Myr of rodent evolution. It is possible, although the supposition is largely untested, that the masticatory apparatus of Paleogene rodents was less efficient than that of some of their extant relatives. Over such a long timescale, natural selection would favour those taxa that are best adapted to their environment and to exploit available dietary resources at any one time. It is not unfeasible that this would result in a more efficient masticatory apparatus if broadly similar environments and resources, i.e. ecological niches, remained relatively stable for long periods. Indeed, gradual increase in molar crown height has been

linked to adaptation to ecological niches over long evolutionary timescales in rodent evolution [54]. It is for this reason that a range of modern rodents were analysed here, representing different specializations of the masticatory apparatus. These modern rodents show configurations of the jaw adductor musculature in which different layers of the masseter complex take their origin on the rostrum. In the earliest rodents, the masseter did not extend onto the rostrum at all (a condition known as protrogomorphy), and instead had its origin only on the ventral surface of the zygomatic arch. The expansion of the masseter onto the rostrum has long been argued to represent an adaptation to increase the efficiency of bite force production, specifically increasing incisor biting efficiency among sciuromorphs [38,55] and molar biting efficiency among hystricomorphs [56]. Therefore, if protrogomorph rodents were included in the analyses conducted in this study they might be expected to have lower bite forces with values closer to those of *Kryptobaatar*, thereby casting doubt on the suggested superiority of early rodents over multituberculates in terms of bite force production. In spite of that, *Kryptobaatar* is less efficient than all the extant rodents examined at all biting positions (including positions that are not necessarily optimized for bite force production, such as molar biting in sciuromorphs and incisor biting in hystricomorphs [18]). Additionally, studies examining the increases of biting efficiency conferred by the rostral origin of masseter muscles have found that the efficiency of incisor biting was increased by only around 5% with sciuromorphy [55] and the efficiency of molar biting by around 10–20% with hystricomorphy [57,58]. Even if rodent bite forces were reduced by this amount *Kryptobaatar* would still largely exhibit lower bite forces than extant rodents. This shows that, despite millions of years of evolution, early rodents with protrogomorph muscle configuration may well have been capable of producing bite forces of similar magnitude to their extant relatives.

Another caveat imparted by supposed adaptation and improved efficiency through evolutionary time is that, despite sharing craniomandibular features with later surviving members of its suborder [34], *Kryptobaatar* dates to 10–35 Myr before the Paleogene multituberculates that were potentially competing with rodents (age range due to dating uncertainties of early rodent, Paleogene multituberculate, and especially *Kryptobaatar* fossil sites, e.g. [59]). During this time window, masticatory efficiency may have evolutionarily increased among multituberculates. An analysis of lower tooth row dental complexity (orientation patch count, OPC) found that *Kryptobaatar* had values (OPC of 132) indicative of animal-dominated omnivory [5]. This value is lower (more towards the carnivorous than herbivorous range of values) than those of all but two extant rodents tested to date, the carnivorous/insectivorous *Hydromys chrysogaster* (OPC = 122) and the animal-dominated omnivore *Akodon serrensis* (OPC = 113), and as such is lower than OPC values for the extant rodents used as comparative taxa in this study (OPC of *Mus* = 167, *Rattus* = 239, *Cavia* = 230) [5,53,60]. This suggests that Paleogene multituberculates with similar values of dental complexity to *Kryptobaatar* were less likely to be in direct dietary competition with contemporaneous Paleogene rodents (for which the majority of available evidence suggests herbivorous and/or granivorous diets, e.g. [61–70]) than were more herbivorous multituberculates with higher OPC values. Given the evolution of more herbivorous diets among multituberculates through the Upper Cretaceous and Paleogene, *Kryptobaatar* may have been less efficient at dealing with tough herbivorous food resources than the presumably better adapted Paleogene multituberculate herbivore specialists (such as the taeniolabidoids). Conducting craniomandibular analyses on taeniolabidoid skulls should be a priority for future research to test this idea, although well-preserved skulls of these Paleocene multituberculates are rare and would require considerable digital restoration.

It has also been suggested that the djadochtatherioidean skull was atypical in several respects due to modifications for burrowing [33] and *Kryptobaatar* does share the wide, flat skull of burrowing, fossorial rodents [71]. The skull of *Kryptobaatar* may therefore have been optimized not only for food processing, but also to deal with additional stresses associated with burrowing and subterranean movement. Extant burrowing rodents are known to exhibit cranial and mandibular morphologies that are distinct from their non-burrowing counterparts [72–76]. Both *in vivo* [77] and FEA [43] studies have shown that such morphological traits facilitate relatively greater bite forces, which enable more effective exploitation of the subterranean niche, particularly in the use of incisors as digging tools. *Kryptobaatar* may have had similarly higher bite forces than non-burrowing multituberculates due to incisor use during burrowing. This hints at the possibility of even greater differences in bite forces between the multituberculates and rodents than found in this study and thus of greater support for the functional superiority of rodents during the Paleogene, but warrants testing in future. It is worth noting that *Kryptobaatar* shares many craniodental features with 'head-lift' digging rodents such as spalacids (blind mole-rats) [72], which use their incisors and spade-shaped skull in combination when burrowing, including: a wide long rostrum, elongate nasals, a deep, flat and wedge-shaped skull,

broad zygomatic arches, with less procumbent incisors than 'chisel-tooth' digging rodents. This suggests that any increase in adductor muscle force associated with incisal digging might be lessened if *Kryptobaatar* used 'head-lift' rather than 'chisel-tooth' digging, since it is not the incisors alone that are involved. Nevertheless, the majority of Asian Upper Cretaceous multituberculates, including *Kryptobaatar*, do not possess unequivocal fossorial adaptations [34,78] and have been reconstructed as terrestrial, moving by saltatory locomotion [34,79,80].

Despite the limitations from model simplification, muscle reconstructions and caveats of comparisons between Cretaceous and extant taxa, this study importantly provides a first step in quantitatively comparing function between multituberculates and rodents, with evidence from bite force estimates lending new support to the CE hypothesis for multituberculate extinction.

## 4.4. Mechanical trade-offs and feeding stresses

Recently, Tseng & Flynn ([81], p. 3) proposed for carnivorans that '[cranial] strength should be just as important as, or more so than, ME [mechanical efficiency] in generalists' to allow processing of a wider variety of food items with variable material properties. It is tempting to suggest that similar selective mechanisms favouring skull strength were operating on many multituberculates, since the majority are reconstructed as omnivores with broader, less plant-based diets than their Paleogene rodent counterparts [5,82]. The low stresses found for *Kryptobaatar*, itself considered an animal-dominated omnivore [5], might be explained in this manner. However, this is unlikely to be the case since the rodents studied here include some of the most successful and pervasive generalist omnivores within the Rodentia, with myomorphs exhibiting substantial foraging plasticity [83]. The myomorphs were, however, more highly stressed than the hard-food specialist sciuromorph that might have been predicted by the Tseng & Flynn [81] model to evolve towards high bite force production at the expense of stress to feed on its diet of hard nuts and seeds [84]. It therefore appears that a generalist omnivorous diet does not require greater skull strength in rodents and is unlikely to explain the low stresses in *Kryptobaatar*.

Given that *Kryptobaatar* has values of dental complexity comparable with extant animal-dominated omnivores [5], and is a small animal (skull length 25–35 mm), fast jaw closure may have been useful in trapping small, fast-moving prey such as insects, as it is among extant insectivores [85,86]. Such fast jaw closure at the expense of bite force has also been reconstructed for the Early Jurassic mammaliaform *Kuehneotherium* in comparison to the contemporaneous *Morganucodon* [31]. Although *Kryptobaatar* produced lower bite forces than the extant rodents overall, lever mechanics predicts that with lower bite forces and lower mechanical advantage the jaws can close with greater speed. This points to a trade-off between maintaining bite force high enough to perhaps only occasionally process resistant plant material while also keeping jaw closure speed high enough to catch fast-moving prey. Herrel *et al.* [87] suggested that just such a trade-off could exist in omnivorous lizards, with the maintenance of high jaw speed to 'grab' insects constraining the evolution of higher bite forces.

The higher stresses seen among sciuromorph and myomorph rodents likely represent a mechanical trade-off between withstanding tolerable craniomandibular stress while maximizing muscle volume to increase bite force. Similar trade-offs have been identified for adaptations to durophagy in extinct carnivorans, where the evolution of high bite force needed to proceed with sufficient protection for the bones of the skull [88]. Additionally, it has been suggested for several mammal groups that selection does not act towards maximizing cranial strength and minimizing stress, but instead towards maximizing mechanical advantage [89,90]. Lower stresses may therefore not indicate competitive superiority if increases in the muscle to bone volume ratio (and hence higher stresses within a tolerable range) were accompanied by other functional advantages, such as production of higher bite forces. Examining the muscle to bone volume ratios in taxa studied here reveals higher ratios among rodents compared to *Kryptobaatar* (see electronic supplementary material), leading to correspondingly higher bite forces.

The tensile and compressive yield stresses of bone have been estimated at 130 and 180 MPa, respectively [91,92]. Although feeding-induced stresses are higher among rodents than *Kryptobaatar*, all taxa studied here operate well below such mechanical limits of bone. McIntosh & Cox [43] recently questioned the value of examining stress in an evolutionary context, since fundamentally it only matters that bone is operating below its yield strength at an appropriate safety factor, and stress variations below this upper limit are likely to be functionally unimportant. Therefore, given the trade-offs discussed above and the fact that only the upper bounds of stress really matter, stress distribution is unlikely to offer great insight into the functional superiority between multituberculates and rodents, and other factors such as optimization for bite force production were probably more selectively important.

## 4.5. Selective advantage of higher bite forces

It is important to assess how higher bite forces may have provided rodents with a competitive advantage, given a possible dominance of granivory in the diets of early rodents and at least some multituberculates [5,44,61], and in the context of the palaeoenvironmental changes that occurred during the Paleogene. As shown for extant rodents and birds, the outer shells of seeds and nuts are unlikely to be broken with a single bite at maximum force, since the yield stresses of these shells are far higher than maximum bite forces [93,94]. Multiple bites are taken to thin the husk before breaking into seeds. The advantage conferred by higher bite forces does allow access to harder foodstuffs but, as suggested by van der Meij & Bout [93] for finches, a perhaps more important advantage comes from a decrease in the time needed to remove the husks of seeds with a hardness below maximum bite force. Higher bite forces would provide selective advantages not only for granivorous taxa, but also for frugivorous, omnivorous, insectivorous and carnivorous taxa. It has been suggested for bats and lizards that bite force has a role in constraining dietary breadth, since the higher bite forces needed to break down harder fruits and to pierce the thicker exoskeletons of larger arthropods may render these resources unavailable to taxa with weaker bites [95–97]. As for seeds, larger and harder animal prey also require more extensive processing over a longer time by animals with lower bite forces [95,97].

Forests were drying during the late Paleocene and early Eocene, when multituberculates were declining at their fastest rates, with soft seeds and fruits becoming scarce, while hard, dry seeds were increasing in prevalence [98–100]. More rapid husking of rarer soft seeds and the ability to crack more common hard seeds may have provided an advantage for rodents over multituberculates in Paleogene environments, and links between the rise of large hard nuts and rise of rodents have been documented previously [61,62,101]. Given the known increases in dietary breadth conferred by greater bite forces, rodents may also have been able to exploit a greater range of other food resources (animals, including insects, as well as plants) than multituberculates during this period. The combination of access to harder foods and more efficient processing of softer items by Paleocene rodents, potentially enabled by higher bite force, provides support for the CE hypothesis. The observed decline among multituberculates through the Paleogene in both mean dental complexity (indicating a general shift away from the herbivore end of the spectrum) and the number of genera reconstructed as plant-dominated omnivores or herbivores [5] hints further at the competitive exclusion of multituberculates from plant-dominated dietary niches by purportedly granivorous Paleocene rodents.

Other aspects of rodent palaeobiology have also been linked to competitive superiority over multituberculates. These include supposed physiological advantages conferred by the eutherian mode of reproduction, with longer gestation and larger neonates compared with multituberculates [34,102,103]. Advantages in locomotion have also been proposed, with multituberculates being apparently incapable of prolonged running [80], which may have allowed rodents to preferentially avoid predation from diversifying Paleogene predators that included strigiform raptors and creodont and carnivoran eutherians [21]. In combination with these other possible factors, the tentative support for the functional superiority of rodents shown in this study now bolsters the long-held view that competition from rodents played a role in the extinction of multituberculates observed in the Paleogene fossil record.

Data accessibility. All accompanying data have been made openly available at the data.bris research data repository (doi:10. 5523/bris.sednr7s6wugv233adbh1j8oyw). CT scan data from DigiMorph for *Kryptobaatar dashzevegi* can also be found at http://digimorph.org/specimens/Kryptobaatar_dashzevegi, for *Mus musculus* at http://www.digimorph.org/ specimens/Mus_musculus and for *Rattus norvegicus* at http://digimorph.org/specimens/Rattus_norvegicus. CT scan data used in the Cox *et al.* studies [18,19,40] is also available on MorphoSource for *Rattus norvegicus* at https:// www.morphosource.org/Detail/SpecimenDetail/show/specimen_id/11777, for *Sciurus carolinensis* at https://www. morphosource.org/Detail/SpecimenDetail/show/specimen_id/11778 and for *Cavia porcellus* at https://www. morphosource.org/Detail/SpecimenDetail/show/specimen_id/11766.

Authors' contributions. P.G.C., S.N.C., I.J.C. and E.J.R. conceived the study. N.F.A., E.J.R. and P.G.C. designed the study approach. N.F.A. collected the data, conducted the biomechanical and statistical analyses, produced the figures and wrote the first draft of the manuscript. All authors interpreted the data, contributed substantially to manuscript revisions, and gave their final approval for publication.

Competing interests. We have no competing interests.

Funding. Data on extant rodents from [18] were originally acquired with support of a grant from the Natural Environment Research Council to P.G.C., Michael J. Fagan (University of Hull) and Nathan S. Jeffery (University of Liverpool) (NE/G001952/1) and to E.J.R. (NE/G001979/1). E.J.R. was also supported by NE/K01496X/1. Grants from The Ella and Georg Ehrnrooth Foundation and the Otto A. Malm Foundation supported the contribution of I.J.C.

Acknowledgements. The authors thank Jessie Maisano and Timothy Rowe (University of Texas at Austin) for enabling access to CT scans from the Digital Morphology online library, and Tom Davies (University of Bristol) for IT

support. For permission to use figures from the entry by R.E. Sloan in Fairbridge and Jablonski, eds., *The Encyclopedia of Paleontology*, we thank David Jablonski, surviving copyright holder. We also thank Julia Schultz and an anonymous reviewer for comments and suggestions that improved the manuscript.

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
