## [Reviewer comments · Royal Society Open Science]

Review History

RSOS-181536.R0 (Original submission)

Review form: Reviewer 1 (Julia Schultz)

Is the manuscript scientifically sound in its present form?

Yes

Are the interpretations and conclusions justified by the results?

Yes

Is the language acceptable?

Yes

Is it clear how to access all supporting data?

Yes

Do you have any ethical concerns with this paper?

No

Have you any concerns about statistical analyses in this paper?

No

Recommendation?

Accept with minor revision (please list in comments)

Comments to the Author(s)

Review: RSOS-1815536

Functional tests of the competitive exclusion hypothesis for multituberculate extinction
by authors

N. Adams, E. Rayfield, P. Cox, S. Cobb, I. Corfe

Summary:

The authors present the very first computer aided testing of the long standing hypothesis that multituberculates were outcompeted by rodents during the Cenozoic. Throughout the manuscript this hypothesis is referred to as “competitive exclusion (CE) hypothesis”. The multituberculate skull morphology and dentition show functional similarities to rodents. Multituberculata were very abundant and diverse in the Mesozoic, but the diversity declined during the mid-late Paleocene and onwards. Multituberculates are often referred to as the “rodents of the Mesozoic” due to the functional similarities. In their study the authors investigate one multituberculate skull and compare to four rodent skulls applying the Finite Element Analysis. The hypotheses being tested are: 1) whether rodent skulls are better adapted to deal with higher stresses linked to different feeding behavior (gnawing versus chewing) and thus show lower stress patterns, 2) rodent skulls can generate higher bite forces than multituberculates, and 3) rodent skulls are more resistant to bending and torsion. In order to address these hypotheses the Upper Cretaceous multituberculate taxon *Kryptobaatar dashzevegi* was compared to extant rodent taxa. The presented results show basically mixed support for the CE hypothesis.

General impression:

The submitted manuscript is well written and well organized, addressing one of the longest standing hypothesis of mammalian evolution. The authors applied adequate methods to address the different hypotheses that were posed. The results are well presented and sufficiently illustrated. In order to clarify certain aspects of the discussion section electronic supplementary material was added. Results are very critically discussed. The manuscript is interesting, because existing weak points of this study are well summarized by authors themselves to a great extent. For example, the choice of taxa is well defended based on existing literature. However, comparing one Upper Cretaceous supposedly omnivorous multituberculate skull to three extant omnivorous rodent taxa and one herbivorous extant rodent taxon still bears problems.

Suggestions for improvement:

In my opinion three main subjects are not addressed:

1) Palinal (multituberculate) versus proal (rodent) chewing movement. The groups evolved opposite directed power strokes in their chewing movements and muscles fibers are operating differently. Thus, proportions and attachment sites of muscles are different, which is also reflected in skull morphology and the different expression of the glenoid fossa. When reading the manuscript the reader gets the impression that both groups do exactly the same, just differ in skull morphology. In order to address this issue, I suggest to present an illustration of the muscle attachment sites for all five taxa. Because the opposing chewing movements exist between rodents and multituberculates, I suggest to also add a map with calculated muscle vectors, for at least one rodent and *Kryptobaatar*. This can be easily added to the electronic supplementary material. This way the paper opens to a broader audience, and is easier to follow for people who

are not working with FEA or anatomy of the mammalian skull.

2) The chosen rodent taxa of this study all possess a significant angular process, multituberculates don't. The angular process is functionally important in rodents as it connects with the medial pterygoid and has influence in chewing performance (see for example Grossnickel 2017 on non-multis). In addition, the coronoid process and position of the jaw joint of multituberculates is fundamentally different from the rodent type. All of these morphological features influence the overall proportions and must have an influence on the FEA analysis. This needs to be addressed at some point in the discussion.

3) The authors mention *Kryptobaatar* is a burrowing animal, which is clearly reflected in the skull morphology. And yet no burrowing rodent taxon was chosen for this study, all are mainly terrestrial even arboreal. The discussion of some statements about borrowing rodents and their similarities to *Kryptobaatar* based on the cited literature will greatly contribute to the manuscript.

Minor issues:

- Some of the presented results are discussed in support for the CE hypothesis, but important aspects that might have had influence apart from chewing performance are neglected (e.g., number of offspring, breeding and nesting behavior, reproduction strategies).
- At one point the authors claim the genus *Kryptobaatar* doesn't change significantly over several million years, but as I understand it this was not tested. This seems a strong statement and needs proof.
- I suggest to add FEA maps for all rodent skulls investigated in this study, not only *Kryptobaatar* versus *Mus*. The authors particularly write in their introduction that they analyzed the data for all rodent types (i.e., sciurognath, hystricomorph etc.). The maps should be added to the electronic supplementary material in order to present the results described and to support the discussion of the hypothesis that rodent skulls are more resistant to bending and torsion. Guinea pig, rat, mouse and squirrel, are mentioned, but no skull shape of these is illustrated. This is necessary, especially when an emphasis is made on the different rodent skull configurations (i.e., sciurognath, hystricomorph etc.).
- Page 6, Lines 130-134: the authors discuss why it's justified to use extant rodent taxa based on their similarities to their earliest ancestors. It is necessary to also assess the main differences here, as they can be used in the discussion below for interpreting some of the major findings of this study.
- The discussion appears generalized and is not addressing specific differences between the investigated rodent taxa. The last paragraph of the discussion ends with interpreting the evolution of mostly herbivorous taxa and how they spread in to different niches during the Cenozoic, however the analysis presented here is mostly based on omnivorous taxa. You can find some additional minor comments within the submitted pdf of the manuscript.

Decision

The study is of great interest to the scientific community, as it is the first computer aided analysis that addresses a major theme of mammalian evolution. I recommend to publish the paper after some modifications are made.

Sincerely,
Julia Schultz.

Universität Bonn, Germany
Steinmann-Institut für Geologie, Mineralogie und Paläontologie

Review form: Reviewer 2

Is the manuscript scientifically sound in its present form?

Yes

Are the interpretations and conclusions justified by the results?

Yes

Is the language acceptable?

Yes

Is it clear how to access all supporting data?

Yes

Do you have any ethical concerns with this paper?

No

Have you any concerns about statistical analyses in this paper?

No

Recommendation?

Accept with minor revision (please list in comments)

Comments to the Author(s)

In this study, Adams and co-authors analyze a multituberculate skull using biomechanical modeling techniques, and compare results with representative extant rodent models. They found that rodents tend to have both higher stresses and higher bite force production capabilities compared to the fossil *Kryptobaatar*. They conclude that superior bite force production in rodents may have been a factor in the clade-level replacement of multituberculates during the Paleogene. The study provides the first test of a functional explanation for the competitive exclusion hypothesis for macroevolutionary trends in the two clades.

The manuscript is well-written, and the modeling protocol fully documented. The authors considered several potential methodological sources of variation in their simulation approach, and provided either validated protocols or sensitivity analyses to verify robustness of their data and statistical comparisons. The discussion and conclusion were made with a clear recognition of the limitations of the particular approach and data. I applaud the authors for sharing all of their data and materials in an open-access repository, truly exemplifying open science.

I have only a few major and minor suggestions for improving the framework of the study:

Major:

-I understand that fossil specimens are difficult to come by for this kind of studies, so there isn't much one can do to increase sample size for fossil species modeled. However, the fact that Rodentia represents one of the most diversified living mammal groups, and the research question is clade-level replacement of multituberculates by rodents, makes the use of only four species models difficult to palate. Could the authors provide a figure or table demonstrating the "representativeness" of the four rodent models used, in terms of how broad of a taxonomic and ecological coverage they represent? In addition, I think it would be beneficial to include some visual comparisons of multituberculate cranial and dental diversity, to provide a firmer context for the hypotheses being tested in the study. Are these two groups morphologically similar enough that a competitive exclusion hypothesis with a functional explanation is the most

appropriate question to ask? I suspect so, but I would like to see a strong case made in the introduction with some direct visual comparisons.

Minor:

-There are several places where there is no space between sentences. Please check.

-P10 L218: how many cross-sectional image stacks were analyzed for section moduli? This information is explained in the supplement, but a summary of the dataset size should be indicated in the main text for context.

-P16 L379: the phrase "progressive adaptation" is a bit troubling to me. I understand that the authors are referring to deep time differences in feeding niche adaptations that may exist between extant rodents and their Paleogene predecessors. However, This phrasing evokes direction and objective to evolutionary change. If there is specific evidence pointing to rodent evolution as broadly progressive (i.e., some trait improving unidirectionally over time), please cite references here. Otherwise, I recommend framing the potential ecological differences between extant and fossil rodents as a product of deep time changes in both environment and species represented (in other words, not necessarily progressive by default).

Decision letter (RSOS-181536.R0)

23-Jan-2019

Dear Mr Adams

On behalf of the Editors, I am pleased to inform you that your Manuscript RSOS-181536 entitled "Functional tests of the competitive exclusion hypothesis for multituberculate extinction" has been accepted for publication in Royal Society Open Science subject to minor revision in accordance with the referee suggestions. Please find the referees' comments at the end of this email.

The reviewers and handling editors have recommended publication, but also suggest some minor revisions to your manuscript. Therefore, I invite you to respond to the comments and revise your manuscript.

- Ethics statement

- Data accessibility

If you wish to submit your supporting data or code to Dryad (<http://datadryad.org/>), or modify your current submission to dryad, please use the following link:
<http://datadryad.org/submit?journalID=RSOS&manu=RSOS-181536>

- **Competing interests**

- **Authors' contributions**

- **Acknowledgements**

- **Funding statement**

Because the schedule for publication is very tight, it is a condition of publication that you submit the revised version of your manuscript before 01-Feb-2019. Please note that the revision deadline will expire at 00.00am on this date. If you do not think you will be able to meet this date please let me know immediately.

When submitting your revised manuscript, you will be able to respond to the comments made by the referees and upload a file "Response to Referees" in "Section 6 - File Upload". You can use this to document any changes you make to the original manuscript. In order to expedite the

processing of the revised manuscript, please be as specific as possible in your response to the referees. We strongly recommend uploading two versions of your revised manuscript:

on behalf of Professor Emily Standen (Associate Editor) and Professor Kevin Padian (Subject Editor)
openscience@royalsociety.org

Associate Editor Comments to Author (Professor Emily Standen):
Dear Adams et al.,

Thank you for submitting your paper to RSOS. We have now received sufficient reviews and are pleased to say they are very positive in their responses. Both reviewers do bring up very important and interesting points for you to consider and that have the potential to increase the impact of your paper significantly.

Both the anatomical and functional differences between groups compared need to be addressed further particularly with respect to skeletal features that will impact the biomechanical forces and performance between groups. In addition the breadth of the species used in the comparison might be improved. The reviewers mention concerns regarding comparing fossils with extant taxa of very differing lifehistory strategies. Although not mentioned by the reviewers, as a non-paleo morphologist I am also interested in what contemporary Rodentia fossils exist that you might be able to include in your comparison with multituberculates for a more accurate understanding of the competition that actually existed between species at the time.

This is a nice contribution to the literature and we look forward to seeing it returned. Please carefully address the comments of both reviewers and resubmit your paper.

Sincerely,

Emily Standen

Reviewer comments to Author:
Reviewer: 1

Comments to the Author(s)

Review: RSOS-1815536

Functional tests of the competitive exclusion hypothesis for multituberculate extinction
by authors

N. Adams, E. Rayfield, P. Cox, S. Cobb, I. Corfe

Summary:

The authors present the very first computer aided testing of the long standing hypothesis that multituberculates were outcompeted by rodents during the Cenozoic. Throughout the manuscript this hypothesis is referred to as "competitive exclusion (CE) hypothesis". The multituberculate skull morphology and dentition show functional similarities to rodents. Multituberculata were very abundant and diverse in the Mesozoic, but the diversity declined during the mid-late Paleocene and onwards. Multituberculates are often referred to as the "rodents of the Mesozoic" due to the functional similarities. In their study the authors investigate one multituberculate skull and compare to four rodent skulls applying the Finite Element Analysis. The hypotheses being tested are: 1) whether rodent skulls are better adapted to deal with higher stresses linked to different feeding behavior (gnawing versus chewing) and thus show lower stress patterns, 2) rodent skulls can generate higher bite forces than multituberculates, and 3) rodent skulls are more resistant to bending and torsion. In order to

address these hypotheses the Upper Cretaceous multituberculate taxon *Kryptobaatar dashzevegi* was compared to extant rodent taxa. The presented results show basically mixed support for the CE hypothesis.

General impression:

The submitted manuscript is well written and well organized, addressing one of the longest standing hypothesis of mammalian evolution. The authors applied adequate methods to address the different hypotheses that were posed. The results are well presented and sufficiently illustrated. In order to clarify certain aspects of the discussion section electronic supplementary material was added. Results are very critically discussed. The manuscript is interesting, because existing weak points of this study are well summarized by authors themselves to a great extent. For example, the choice of taxa is well defended based on existing literature. However, comparing one Upper Cretaceous supposedly omnivorous multituberculate skull to three extant omnivorous rodent taxa and one herbivorous extant rodent taxon still bears problems.

Suggestions for improvement:

In my opinion three main subjects are not addressed:

- 1) Palinal (multituberculate) versus proal (rodent) chewing movement. The groups evolved opposite directed power strokes in their chewing movements and muscles fibers are operating differently. Thus, proportions and attachment sites of muscles are different, which is also reflected in skull morphology and the different expression of the glenoid fossa. When reading the manuscript the reader gets the impression that both groups do exactly the same, just differ in skull morphology. In order to address this issue, I suggest to present an illustration of the muscle attachment sites for all five taxa. Because the opposing chewing movements exist between rodents and multituberculates, I suggest to also add a map with calculated muscle vectors, for at least one rodent and *Kryptobaatar*. This can be easily added to the electronic supplementary material. This way the paper opens to a broader audience, and is easier to follow for people who are not working with FEA or anatomy of the mammalian skull.
- 2) The chosen rodent taxa of this study all possess a significant angular process, multituberculates don't. The angular process is functionally important in rodents as it connects with the medial pterygoid and has influence in chewing performance (see for example Grossnickel 2017 on non-multis). In addition, the coronoid process and position of the jaw joint of multituberculates is fundamentally different from the rodent type. All of these morphological features influence the overall proportions and must have an influence on the FEA analysis. This needs to be addressed at some point in the discussion.
- 3) The authors mention *Kryptobaatar* is a burrowing animal, which is clearly reflected in the skull morphology. And yet no burrowing rodent taxon was chosen for this study, all are mainly terrestrial even arboreal. The discussion of some statements about borrowing rodents and their similarities to *Kryptobaatar* based on the cited literature will greatly contribute to the manuscript.

Minor issues:

- Some of the presented results are discussed in support for the CE hypothesis, but important aspects that might have had influence apart from chewing performance are neglected (e.g., number of offspring, breeding and nesting behavior, reproduction strategies).
- At one point the authors claim the genus *Kryptobaatar* doesn't change significantly over several million years, but as I understand it this was not tested. This seems a strong statement and needs proof.
- I suggest to add FEA maps for all rodent skulls investigated in this study, not only *Kryptobaatar* versus *Mus*. The authors particularly write in their introduction that they analyzed the data for all rodent types (i.e., sciurognath, hystricomorph etc.). The maps should be added to the electronic supplementary material in order to present the results described and to support the

discussion of the hypothesis that rodent skulls are more resistant to bending and torsion. Guinea pig, rat, mouse and squirrel, are mentioned, but no skull shape of these is illustrated. This is necessary, especially when an emphasis is made on the different rodent skull configurations (i.e., sciurognath, hystricomorph etc.).

- Page 6, Lines 130-134: the authors discuss why it's justified to use extant rodent taxa based on their similarities to their earliest ancestors. It is necessary to also assess the main differences here, as they can be used in the discussion below for interpreting some of the major findings of this study.

- The discussion appears generalized and is not addressing specific differences between the investigated rodent taxa. The last paragraph of the discussion ends with interpreting the evolution of mostly herbivorous taxa and how they spread in to different niches during the Cenozoic, however the analysis presented here is mostly based on omnivorous taxa.

You can find some additional minor comments within the submitted pdf of the manuscript.

Decision

The study is of great interest to the scientific community, as it is the first computer aided analysis that addresses a major theme of mammalian evolution. I recommend to publish the paper after some modifications are made.

Sincerely,
Julia Schultz.

Universität Bonn, Germany
Steinmann-Institut für Geologie, Mineralogie und Paläontologie

Reviewer: 2

Comments to the Author(s)

In this study, Adams and co-authors analyze a multituberculate skull using biomechanical modeling techniques, and compare results with representative extant rodent models. They found that rodents tend to have both higher stresses and higher bite force production capabilities compared to the fossil *Kryptobaatar*. They conclude that superior bite force production in rodents may have been a factor in the clade-level replacement of multituberculates during the Paleogene. The study provides the first test of a functional explanation for the competitive exclusion hypothesis for macroevolutionary trends in the two clades.

The manuscript is well-written, and the modeling protocol fully documented. The authors considered several potential methodological sources of variation in their simulation approach, and provided either validated protocols or sensitivity analyses to verify robustness of their data and statistical comparisons. The discussion and conclusion were made with a clear recognition of the limitations of the particular approach and data. I applaud the authors for sharing all of their data and materials in an open-access repository, truly exemplifying open science.

I have only a few major and minor suggestions for improving the framework of the study:

Major:

-I understand that fossil specimens are difficult to come by for this kind of studies, so there isn't much one can do to increase sample size for fossil species modeled. However, the fact that

Rodentia represents one of the most diversified living mammal groups, and the research question is clade-level replacement of multituberculates by rodents, makes the use of only four species models difficult to palate. Could the authors provide a figure or table demonstrating the "representativeness" of the four rodent models used, in terms of how broad of a taxonomic and ecological coverage they represent? In addition, I think it would be beneficial to include some visual comparisons of multituberculate cranial and dental diversity, to provide a firmer context for the hypotheses being tested in the study. Are these two groups morphologically similar enough that a competitive exclusion hypothesis with a functional explanation is the most appropriate question to ask? I suspect so, but I would like to see a strong case made in the introduction with some direct visual comparisons.

Minor:

-There are several places where there is no space between sentences. Please check.

-P10 L218: how many cross-sectional image stacks were analyzed for section moduli? This information is explained in the supplement, but a summary of the dataset size should be indicated in the main text for context.

-P16 L379: the phrase "progressive adaptation" is a bit troubling to me. I understand that the authors are referring to deep time differences in feeding niche adaptations that may exist between extant rodents and their Paleogene predecessors. However, This phrasing evokes direction and objective to evolutionary change. If there is specific evidence pointing to rodent evolution as broadly progressive (i.e., some trait improving unidirectionally over time), please cite references here. Otherwise, I recommend framing the potential ecological differences between extant and fossil rodents as a product of deep time changes in both environment and species represented (in other words, not necessarily progressive by default).

Author's Response to Decision Letter for (RSOS-181536.R0)

See Appendix A.

Decision letter (RSOS-181536.R1)

21-Feb-2019

Dear Mr Adams,

I am pleased to inform you that your manuscript entitled "Functional tests of the competitive exclusion hypothesis for multituberculate extinction" is now accepted for publication in Royal Society Open Science.

Royal Society Open Science operates under a continuous publication model (<http://bit.ly/cpFAQ>). Your article will be published straight into the next open issue and this

will be the final version of the paper. As such, it can be cited immediately by other researchers. As the issue version of your paper will be the only version to be published I would advise you to check your proofs thoroughly as changes cannot be made once the paper is published.

on behalf of Professor Emily Standen (Associate Editor) and Professor Kevin Padian (Subject Editor)
openscience@royalsociety.org

Appendix A

Centre for Palaeobiology Research
School of Geography, Geology and the Environment
University of Leicester, Leicester, LE1 7RH, UK

19 February 2019

Department of Biology
University of Ottawa, Ottawa, K1N 6N5, Canada

Submissions of revisions for accepted ms. no. RSOS-181536

Dear Prof. Emily Standen,

I am submitting the revisions to ms. no. RSOS-181536 (Functional tests of the competitive exclusion hypothesis for multituberculate extinction) for your consideration for publication in Royal Society Open Science. I would like to thank you and both the reviewers for constructive comments that have improved the manuscript.

For our point-by-point response to your comments and the comments of the reviewers, please see the pages following this letter (our responses in red text). Line numbers referred to in our responses refer to the track-changed manuscript.

Written permissions for re-use of images from other published sources in the new Figures 1 and 2 have been secured and all image sources and their publishers are fully acknowledged in the captions. Evidence of these permissions can be provided upon request:

- The American Museum of Natural History has granted permission to re-use images in Figures 1a, 1d, 2a and 2f.
- The University of Wyoming has granted permission to re-use images in Figure 1e.
- The American Philosophical Society has granted permission to re-use images in Figures 1f, 1g, 1i, 2b, 2c and 2e.
- Springer Nature has granted permissions to re-use images in Figure 2d.
- Figures 1b and 1h do not require permission as they were published in *Acta Palaeontologica Polonica* under a CC-BY licence (see <https://www.app.pan.pl/copyright-policy.html>).
- The image of the lower jaw in Figure 1c comes under the “fair use” permission of the Geological Society of America for using a single figure from a GSA publication (see https://www.geosociety.org/GSA/Publications/Info_Services/Copyright/GSA/Pubs/guide/copyright.aspx).
- The publishers of the original volume containing the image of the cranium in Figure 1c (Dowden, Hutchinson & Ross, Inc.) are no longer an active company, so could not be contacted for permission. I

contacted the publisher of the e-book version of the publication (Springer Nature), who suggested that permission from the editors of the original volume containing the re-used image would provide sufficient permissions for re-use. Written permission from the sole living editor of the volume, Professor David Jablonski, has been secured. Acting on advice from David Jablonski, his permission is included in the Acknowledgements section rather than the caption.

Thank you for your consideration of this revised manuscript.

Sincerely,
Neil Adams

Mr. Neil F. Adams BSc (Hons) MSc AFHEA
E-mail: nfa10@leicester.ac.uk
Tel: +44(0) 7516 841341

Author response to editor and reviewer comments

Line numbers mentioned in our responses refer to the track-changed manuscript

Associate Editor Comments to Author (Professor Emily Standen):

Dear Adams et al.,

Thank you for submitting your paper to RSOS. We have now received sufficient reviews and are pleased to say they are very positive in their responses. Both reviewers do bring up very important and interesting points for you to consider and that have the potential to increase the impact of your paper significantly.

Both the anatomical and functional differences between groups compared need to be addressed further particularly with respect to skeletal features that will impact the biomechanical forces and performance between groups. In addition the breadth of the species used in the comparison might be improved. The reviewers mention concerns regarding comparing fossils with extant taxa of very differing lifehistory strategies. Although not mentioned by the reviewers, as a non-paleo morphologist I am also interested in what contemporary Rodentia fossils exist that you might be able to include in your comparison with multituberculates for a more accurate understanding of the competition that actually existed between species at the time.

The points raised by the reviewers are addressed below. In addition, contemporaneous Paleogene rodents are considered more fully in the comparison with extant rodents (lines 135-148) and examples are now illustrated in the new Figure 2.

This is a nice contribution to the literature and we look forward to seeing it returned. Please carefully address the comments of both reviewers and resubmit your paper.

Sincerely,

Emily Standen

Reviewer 1

Summary:

The authors present the very first computer aided testing of the long standing hypothesis that multituberculates were outcompeted by rodents during the Cenozoic. Throughout the manuscript this hypothesis is referred to as “competitive exclusion (CE) hypothesis”. The multituberculate skull morphology and dentition show functional similarities to rodents. Multituberculata were very abundant and diverse in the Mesozoic, but the diversity declined during the mid-late Paleocene and onwards. Multituberculates are often referred to as the “rodents of the Mesozoic” due to the functional similarities. In their study the authors investigate one multituberculate skull and compare to four rodent skulls applying the Finite Element Analysis. The hypotheses being tested are: 1) whether rodent skulls are better adapted to deal with higher stresses linked to different feeding behavior (gnawing versus chewing) and thus show lower stress patterns, 2) rodent skulls can generate higher bite forces than multituberculates, and 3) rodent skulls are more resistant to bending and torsion. In order to address these hypotheses the Upper Cretaceous multituberculate taxon *Kryptobaatar dashzevegi* was compared to extant rodent taxa. The presented results show basically mixed support for the CE hypothesis.

General impression:

The submitted manuscript is well written and well organized, addressing one of the longest standing hypotheses of mammalian evolution. The authors applied adequate methods to address the different hypotheses that were posed. The results are well presented and sufficiently illustrated. In order to clarify certain aspects of the discussion section electronic supplementary material was added. Results are very critically discussed. The manuscript is interesting, because existing weak points of this study are well summarized by authors themselves to a great extent. For example, the choice of taxa is well defended based on existing literature. However, comparing one Upper Cretaceous supposedly omnivorous multituberculate skull to three extant omnivorous rodent taxa and one herbivorous extant rodent taxon still bears problems.

Suggestions for improvement:

In my opinion three main subjects are not addressed:

1) Palinal (multituberculate) versus proal (rodent) chewing movement. The groups evolved opposite directed power strokes in their chewing movements and muscle fibers are operating differently. Thus, proportions and attachment sites of muscles are different, which is also reflected in skull morphology and the different expression of the glenoid fossa. When reading the manuscript the reader gets the impression that both groups do exactly the same, just differ in skull morphology. In order to address this issue, I suggest to present an illustration of the muscle attachment sites for all five taxa. Because the opposing chewing movements exist between rodents and multituberculates, I suggest to also add a map with calculated muscle vectors, for at least one

rodent and *Kryptobaatar*. This can be easily added to the electronic supplementary material. This way the paper opens to a broader audience, and is easier to follow for people who are not working with FEA or anatomy of the mammalian skull.

The different chewing movements of multituberculates and rodents are now addressed with:

- A new Figure 3 showing muscle attachment sites and vectors for *Kryptobaatar* and *Mus*.
- A new Figure S1 showing 3D muscle reconstruction for *Kryptobaatar* for comparison with 3D muscle proportions of rodents in already published works (e.g., Baverstock *et al.*, Cox *et al.*)
- A new Figure S2 showing muscle attachment sites for the other three rodents.
- A new section (b) of the Discussion on 'Masticatory and musculoskeletal differences between multituberculates and rodents'.

2) The chosen rodent taxa of this study all possess a significant angular process, multituberculates don't. The angular process is functionally important in rodents as it connects with the medial pterygoid and has influence in chewing performance (see for example Grossnickel 2017 on non-multis). In addition, the coronoid process and position of the jaw joint of multituberculates is fundamentally different from the rodent type. All of these morphological features influence the overall proportions and must have an influence on the FEA analysis. This needs to be addressed at some point in the discussion.

The implications of differently structured morphological features (such as the glenoid fossa, condylar and coronoid processes, angular process, and pterygoid shelf) are now all addressed in the new section (b) of the Discussion on 'Masticatory and musculoskeletal differences between multituberculates and rodents'.

3) The authors mention *Kryptobaatar* is a burrowing animal, which is clearly reflected in the skull morphology. And yet no burrowing rodent taxon was chosen for this study, all are mainly terrestrial even arboreal. The discussion of some statements about burrowing rodents and their similarities to *Kryptobaatar* based on the cited literature will greatly contribute to the manuscript.

Discussion of burrowing rodents and similarities to *Kryptobaatar* have been added (lines 503-529).

Minor issues:

- Some of the presented results are discussed in support for the CE hypothesis, but important aspects that might have had influence apart from chewing performance are neglected (e.g., number of offspring, breeding and nesting behavior, reproduction strategies).

Consideration of other factors relevant for the CE hypothesis, such as reproduction but also locomotion, has now been added to the discussion (lines 633-640).

- At one point the authors claim the genus *Kryptobaatar* doesn't change significantly over several million years, but as I understand it this was not tested. This seems a strong statement and needs proof.

We do not claim this at any point in the manuscript and do not know where the reviewer came across this idea in the text.

It possibly arose from the sentence "*Kryptobaatar* dates to 10-35 million years before the Paleogene multituberculates that were potentially competing with rodents." The time range of '10-35 million years' relates to dating uncertainties of the *Kryptobaatar*, early rodent and Paleogene multituberculate fossil sites rather than a known 10-35 million-year persistence of the genus *Kryptobaatar* in unchanged form. Added "age range due to dating uncertainties of early rodent, Paleogene multituberculate, and especially *Kryptobaatar* fossil sites, e.g. [50]" [Kielan-Jaworowska et al. (2003)] in lines 480-481 to clarify.

- I suggest to add FEA maps for all rodent skulls investigated in this study, not only *Kryptobaatar* versus *Mus*. The authors particularly write in their introduction that they analyzed the data for all rodent types (i.e., sciurognath, hystricomorph etc.). The maps should be added to the electronic supplementary material in order to present the results described and to support the discussion of the hypothesis that rodent skulls are more resistant to bending and torsion. Guinea pig, rat, mouse and squirrel, are mentioned, but no skull shape of these is illustrated. This is necessary, especially when an emphasis is made on the different rodent skull configurations (i.e., sciurognath, hystricomorph etc.).

We do not claim to have undertaken FEA on the crania of all rodent types (sciurognath, hystricomorph, myomorph) in the Introduction. As we mentioned explicitly in section (b) of the Material and methods in the submitted manuscript, "digital models and results from cranial FEA of three extant rodents (*Sciurus*, *Cavia*, *Rattus*) were taken directly from the Cox et al. study and were used for comparison with the new results for the *Kryptobaatar* cranium calculated herein". We show the cranial FEA results for *Kryptobaatar* and *Mus*, because these are the new results from our study. The cranial FEA for *Sciurus*, *Cavia* and *Rattus* was done in a previous study and so we do not show the FEA maps because they are already published elsewhere.

The skull shape of the mouse is shown in figure 4b and 4d. The FEA maps from the Cox et al. (2012) study have been added to the supplementary material (figure S12) to make it easier for readers to directly compare our results with the previously published models. These FEA maps also illustrate the skull shape for the guinea pig, rat and squirrel as requested. Skull shapes for the rat, squirrel and guinea pig are also illustrated in the new Figure 2, comparing them to Paleogene rodent skulls.

- Page 6, Lines 130-134: the authors discuss why it's justified to use extant rodent taxa based on their similarities to their earliest ancestors. It is necessary to also assess the main differences here, as they can be used in the discussion below for interpreting some of the major findings of this study.

The differences between extant and extinct forms have been added and similarities have been revised and expanded (see lines 135-148).

- The discussion appears generalized and is not addressing specific differences between the investigated rodent taxa.

The specific differences between investigated rodent taxa are not the focus of this paper and have been discussed at length by Cox *et al.* (2012, PLoS ONE, 7: e36299). The focus of this paper is on the comparison between rodents and the multituberculate *Kryptobaatar*. In section (a) of the Discussion we do discuss the specific differences between different rodents and *Kryptobaatar* and how these relate to our initial hypotheses.

The last paragraph of the discussion ends with interpreting the evolution of mostly herbivorous taxa and how they spread in to different niches during the Cenozoic, however the analysis presented here is mostly based on omnivorous taxa.

We have expanded this discussion to note that the benefits of higher bite forces are not just applicable to granivory, but also to frugivory, omnivory, insectivory and carnivory (lines 607-614 and 621-624). As discussed in the manuscript, the taxa analysed are imperfect representatives of Paleocene rodents and Paleocene multituberculates, so we do extend inferences based on omnivorous taxa to purportedly herbivorous/granivorous taxa. We discuss these caveats in the manuscript (discussion section (c) of the revised manuscript) and eagerly await the testing of our results and conclusions in future work.

You can find some additional minor comments within the submitted pdf of the manuscript [copied below from the annotated pdf]:

1) Keywords: delete 'competitive exclusion' – This is already part of the title of the manuscript

Deleted and replaced with 'macroevolution'.

2) Line 29 "four extant rodents": even though well discussed, I think this bears some problems. See my comments in the letter.

Addressed above.

3) Lines 132-134: I suggest to not only cite the similarities, but add also main differences between early and advanced. This way it might be easier to circle back in the discussion.

Differences added.

4) Line 299: Please add the FEA maps of all rodents you investigated in the supplement. Also muscle attachment maps would help the reader to follow.

As mentioned above, cranial FEA maps for the squirrel, guinea pig and rat have now been added to the supplement to enable easy comparison, but they are not new results from this study. Muscle attachment maps added in Figures 3 and S2.

5) Lines 388-392: Assumption, this needs some support from literature.

The preceding lines of the manuscript (lines 457-460) reference key literature comparing masticatory efficiency of protrogomorph rodents with sciurormorph and hystricomorph rodents. Additional reference to Druzinsky (2010), which demonstrates lower biting efficiency at the incisors in prototrogomorph vs sciurormorph rodents, added in line 459.

Also, the sentence “Therefore, if protrogomorph rodents were included in the analyses conducted in this study they might be expected to have lower bite forces with values closer to those of *Kryptobaatar*...” is a prediction rather than an assumption. It is something to be tested by future work, not something we assume to be true.

6) Lines 401-402: Assumption, this needs some support from literature

The preceding lines of the manuscript (lines 464-472) provide literature support for our suggestion that early Paleogene protrogomorph rodents could produce similar bite forces to extant rodents. We have added “with protrogomorph muscle configuration” in line 473-474 to highlight the relevance of the preceding literature for our suggestion.

7) Lines 513-514: But *Kryptobaatar* is assumed to have been omnivorous

Addressed above.

8) Line 529: I suggest to add the links for the data used from DigiMorph here, too.

Links to DigiMorph data have been added, as well as links to MorphoSource for rodent CT scan data.

9) Figure 2: Main issue here is the presence of the angular process in all rodents. *Kryptobaatar* has none, so the von Mises stress pattern have to be different.

This point is addressed in the new section (b) of the Discussion. Despite lacking an angular process, multituberculates have an extensive pterygoid shelf that is analogous in several ways (including adductor muscle attachment).

Decision:

The study is of great interest to the scientific community, as it is the first computer aided analysis that addresses a major theme of mammalian evolution. I recommend to publish the paper after some modifications are made.

Sincerely,
Julia Schultz.

Universität Bonn, Germany
Steinmann-Institut für Geologie, Mineralogie und Paläontologie

Reviewer 2

In this study, Adams and co-authors analyze a multituberculate skull using biomechanical modeling techniques, and compare results with representative extant rodent models. They found that rodents tend to have both higher stresses and higher bite force production capabilities compared to the fossil *Kryptobaatar*. They conclude that superior bite force production in rodents may have been a factor in the clade-level replacement of multituberculates during the Paleogene. The study provides the first test of a functional explanation for the competitive exclusion hypothesis for macroevolutionary trends in the two clades.

The manuscript is well-written, and the modeling protocol fully documented. The authors considered several potential methodological sources of variation in their simulation approach, and provided either validated protocols or sensitivity analyses to verify robustness of their data and statistical comparisons. The discussion and conclusion were made with a clear recognition of the limitations of the particular approach and data. I applaud the authors for sharing all of their data and materials in an open-access repository, truly exemplifying open science.

I have only a few major and minor suggestions for improving the framework of the study:

Major:

-I understand that fossil specimens are difficult to come by for this kind of studies, so there isn't much one can do to increase sample size for fossil species modeled. However, the fact that Rodentia represents one of the most diversified living mammal groups, and the research question is clade-level replacement of multituberculates by rodents, makes the use of only four species models difficult to palate. Could the authors provide a figure or table demonstrating the "representativeness" of the four rodent models used, in terms of how broad of a taxonomic and ecological coverage they represent? In addition, I think it would be beneficial to include some visual comparisons of multituberculate cranial and dental diversity, to provide a firmer context for the hypotheses being tested in the study. Are these two groups morphologically similar enough that a competitive exclusion hypothesis with a functional explanation is the most appropriate question to ask? I suspect so, but I would like to see a strong case made in the introduction with some direct visual comparisons.

Two new figures have been produced as requested.

The first (Figure 1) provides an indication of craniodental diversity among some of the major Paleogene families of multituberculates (Ptilodontidae, Neoplagiaulacidae, Eucosmodontidae, Microcosmodontidae, Taeniolabididae, Lambdopsalidae) compared to *Kryptobaatar*. Relatively complete skulls of Paleogene multituberculates are known only from four groups (Ptilodontidae,

Neoplagiulacidae, Taeniolabididae, Lambdopsalidae) as far as the authors are aware.

Although our research question is of clade-level replacement and rodents are indeed extremely diverse today, during the Paleocene and early Eocene rodents were not nearly as diverse. Rather than attempt to highlight how the extant rodents represent the taxonomic and ecological coverage of all modern rodents, Figure 2 now provides a comparison of how the modern rodents compare to the Paleogene rodents that would have been in competition with multituberculates (as requested by the Associate Editor), which we believe is more relevant to our research question.

Minor:

-There are several places where there is no space between sentences. Please check.

Spaces added where they were missing.

-P10 L218: how many cross-sectional image stacks were analyzed for section moduli? This information is explained in the supplement, but a summary of the dataset size should be indicated in the main text for context.

Details of the procedures for calculating section moduli and polar moments of inertia from the image stacks have been added in lines 234-237.

-P16 L379: the phrase "progressive adaptation" is a bit troubling to me. I understand that the authors are referring to deep time differences in feeding niche adaptations that may exist between extant rodents and their Paleogene predecessors. However, This phrasing evokes direction and objective to evolutionary change. If there is specific evidence pointing to rodent evolution as broadly progressive (i.e., some trait improving unidirectionally over time), please cite references here. Otherwise, I recommend framing the potential ecological differences between extant and fossil rodents as a product of deep time changes in both environment and species represented (in other words, not necessarily progressive by default).

Rephrased with reference to selection acting on taxa present at any one time that are best adapted to their environment, including reference to a study (Tapaltsyán *et al.*, 2015, Cell Rep. 11: 673-680) describing relatively progressive increase in rodent tooth crown height over millions of years – see lines 443-450.